# Meta-VBO: Utilizing Prior Tasks in Optimizing Risk Measures with Gaussian Processes

**Quoc Phong Nguyen[1], Bryan Kian Hsiang Low[2] & Patrick Jaillet[1]**
[1]LIDS and EECS, Massachusetts Institute of Technology, USA
[2]School of Computing, National University of Singapore, Singapore
`qphongmp@gmail.com`, `lowkh@comp.nus.edu.sg`, `jaillet@mit.edu`

## Abstract

Research on optimizing the risk measure of a blackbox function using Gaussian processes, especially *Bayesian optimization* (BO) of risk measures, has become increasingly important due to the inevitable presence of uncontrollable variables in real-world applications. Nevertheless, existing works on BO of risk measures start the optimization from scratch for every new task without considering the results of prior tasks. In contrast, its vanilla BO counterpart has received a thorough investigation on utilizing prior tasks to speed up the current task through the body of works on *meta-BO* which, however, have not considered risk measures. To bridge this gap, this paper presents the first algorithm for *meta-BO of risk measures* (i.e., *value-at-risk* (VaR) and the *conditional VaR*), namely meta-VBO, by introducing a novel adjustment to the upper confidence bound acquisition function. Our proposed algorithm exhibits two desirable properties: (i) invariance to scaling and vertical shifting of the blackbox function and (ii) robustness to prior harmful tasks. We provide a theoretical performance guarantee for our algorithm and empirically demonstrate its performance using several synthetic function benchmarks and real-world objective functions.

## 1 Introduction

Real world optimization problems, such as maximizing the performance of a manufacturing line, the crop yield of a farm, and the performance of *machine learning* (ML) models, often face with the challenge of dealing with blackbox objective functions (i.e., without closed-form expressions, convexity, or derivatives) and time-consuming or expensive noisy zeroth-order evaluations (Snoek et al., 2012; Nguyen et al., 2021b). These issues have attracted significant attention, particularly on the *Bayesian optimization* (BO) framework (Srinivas et al., 2010; Snoek et al., 2012; Shahriari et al., 2015; Hernández-Lobato et al., 2014; Wang and Jegelka, 2017; Garnett, 2022). Recently, there has been an increasing interest in *BO of risk measures*, driven by the need to address practical aspects related to uncontrollable variables referred to as *environmental random variables* (Cakmak et al., 2020), e.g., random errors in sensors of the manufacturing line, weather conditions in farms (Nguyen et al., 2021b;a), and random perturbation in training data of ML models (Bogunovic et al., 2018). BO of risk measure algorithms specifically search for the controllable *decision variables* (e.g., applied torque, input force, nutrient amounts, and ML model hyperparameters) that optimize a *risk measure* of the blackbox function where the risk is induced by the environmental random variables. Risk measures investigated in existing works include the worst-case value (Bogunovic et al., 2018), *value-at-risk* (VaR) and *conditional VaR* (CVaR) (Cakmak et al., 2020; Nguyen et al., 2021b;a; Picheny et al., 2022), probability threshold robustness (Iwazaki et al., 2021a), and the variance (Iwazaki et al., 2021b; Makarova et al., 2021).

While there is an extensive research on BO of risk measures, using learning experiences from *prior tasks* to accelerate the optimization of the *current task* has not been explored. In contrast, this approach has been shown to effectively reduce the number of costly observations required by the *vanilla BO* counterpart through numerous works on *meta-BO*. These works encompass a wide spectrum of methodologies, ranging from those that involve the transfer of the probabilistic models from prior tasks to the current one (Swersky et al., 2013), to more nuanced approaches centered on transferring the optimization result (Feurer et al., 2018; Wistuba et al., 2016; 2018; Perrone et al.,

2019; Ramachandran et al., 2019; Volpp et al., 2020; Dai et al., 2022). Although the former category, such as the multi-task *Gaussian processes* (GP) (Swersky et al., 2013), is well-defined, it often suffers from computational challenges as discussed by Dai et al. (2022) and is not specifically tailored to the goal of optimization - the blackbox function's maximizer. In contrast, the latter category, which attracts the majority of research attention, presents greater challenges and lacks a clearly defined framework. For example, Feurer et al. (2018) transfers the ranking of function evaluations. Perrone et al. (2019) narrows down the search space. Ramachandran et al. (2019) considers the distribution of the optima. Dai et al. (2022) considers a weighted average of the acquisition functions.

In this paper, we focus on the challenge of transferring the optimization outcome. Specifically, we pioneer the first solution to the *meta-BO of risk measure* problem, namely *meta-VBO*. Our work accounts for the risk introduced by the environmental random variables - a crucial aspect not addressed in any existing meta-BO works. We focus on two well-known risk measures: *value-at-risk* (VaR) and *conditional VaR* (CVaR) (Cakmak et al., 2020; Torossian et al., 2020). Moreover, our solution also extends to the adversarially robust BO (Bogunovic et al., 2018) and applies to the meta-BO problem. We assume that the distribution of the environmental random variables is given, e.g., by estimating from historical data or domain knowledge. Additionally, we assume that during the optimization, noisy evaluations of the blackbox function can be acquired (queried) at any realizations of the environmental random variables. In the above examples, this can be achieved running simulations of manufacturing lines with controlled random errors or growing indoor plants in a laboratory environment where weather conditions are regulated. After the optimization is completed, we recommend implementing the best decision variable in the real-world environment where the environmental random variables are beyond our control. [1] These assumptions are commonly made in several BO works with risk measures (Cakmak et al., 2020; Nguyen et al., 2021b;a).

Under the above settings, there are several major challenges in addressing meta-BO of risk measures. **First**, the choice of the information to be transferred from *prior tasks* to the *current task* should be tailored to the optimization goal of determining the optima's locations. Since they are invariant under scaling and vertical-shifting (i.e., increasing or decreasing the function by a constant) of the blackbox function, it is crucial to ensure that the proposed meta-BO of risk measure solution maintains this invariance. Hence, it rules out the use of the function gap (i.e., the maximum difference between function evaluations) in the theoretical study of Dai et al. (2022) as a similarity measure between tasks. **Second**, as the number of prior tasks grows, the possibility of encountering a task that hampers the current optimization task arises (Feurer et al., 2018; Dai et al., 2022). Thus, it is essential to be resilient against such prior harmful tasks.

Moreover, one cannot easily extend existing meta-BO solutions to address the BO problem with risk measures. While the posterior belief of the blackbox function follows a GP in the vanilla BO, many risk measures including VaR and CVaR do not follow any well-known stochastic process. Hence, it is challenging to adopt approaches based on the Gaussian posterior distribution in the works of Ramachandran et al. (2019) and Dai et al. (2022). Furthermore, risk measures are not directly observed. They are only estimated from noisy evaluations of the blackbox function. As a result, although Feurer et al. (2018) is able to discover and assign low weights to prior harmful tasks by proposing a ranking loss function, extending this loss function to tackle risk measures is not immediately obvious. Last, there are not any existing meta-BO works that possess both theoretical performance guarantees and robustness against scaling and vertical-shifting of the blackbox function and prior harmful tasks.

To address the above problems, we introduce a new perspective in constructing our solution, namely *meta-VBO*. Specifically, while the classic *Gaussian process - upper confidence bound* (GP-UCB) algorithm (Srinivas et al., 2010) offers only one option of the input query at each BO iteration, we pursue a different approach of constructing a *versatile query set* (V-set) in Sec. 3.1. Interestingly, we can obtain a no-regret algorithm by selecting any input in this set as the input query (Theorem 3.2). To the best of our knowledge, this characteristic has not been introduced in any BO algorithm. On one hand, it implies that by restricting the choice of the input query to V-set, the algorithm is no-regret regardless of prior harmful tasks. On the other hand, the flexibility in choosing the input query from a set (as opposed to one option in GP-UCB) allows us to transfer information from prior tasks to the current task by assigning different preferences/priorities to inputs in V-set (Sec. 3.2). Essentially, inputs that are likely to be local (or global) maximizers of many prior tasks are assigned high priorities.

---

[1] The notion of the best decision variable is discussed in Sec. 3.3.

The size of V-set can be tuned via two hyperparameters, which allows a trade-off between the amount of information to be incorporated from prior tasks and the cumulative regret bound of the current ask. This reflects the intuition that without knowing the similarity between prior tasks and the current task, utilizing more information from prior tasks may not benefit the current task. We justify the selected hyperparameters in the experiments based on the incurred regret. Furthermore, V-set is adaptively updated as observations are accumulated from one iteration to the next. This naturally adjusts the effect of prior tasks on the current task as discussed in Remark 3.4, ensuring meta-VBO's robustness to prior harmful tasks. In Sec. 3.3, we propose a strategy to recommend an input that approximates the optimal solution at each iteration such that its instantaneous regret diminishes towards zero as the iteration progresses (Theorem 3.7). Last, we empirically evaluate the performance of meta-VBO in optimizing VaR and CVaR of several synthetic and real-world objective functions in Sec. 4.

## 2 PRELIMINARY

### 2.1 BO OF RISK MEASURES

We are interested in maximizing a blackbox objective function $f$ (e.g., the performance of a system) that depends on both the *decision variable* $\mathbf{x} \in \mathcal{X}$ and the *environmental variable* $\mathbf{z} \in \mathcal{Z}$, where $\mathcal{X}$ and $\mathcal{Z}$ are bounded subsets of $\mathbb{R}^d$ and $\mathbb{R}^{d'}$, respectively. We assume that $\mathcal{X}$ and $\mathcal{Z}$ are discrete for simplicity, though the results can be extended to continuous domains (Srinivas et al., 2010; Chowdhury and Gopalan, 2017). Furthermore, in the real-world environment, $\mathbf{x}$ is controllable while $\mathbf{z}$ (e.g., weather conditions and random errors) is not. Hence, the environmental variable is represented as a random variable, denoted as $\mathbf{Z}$. We assume that the distribution of $\mathbf{Z}$ is known, e.g., by estimating from historical data or domain knowledge. These settings are adopted from the existing works of Cakmak et al. (2020); Nguyen et al. (2021b;a).

Due to the randomness in $\mathbf{Z}$, evaluating the blackbox function at a particular value of $\mathbf{x}$ can result in a high value under one realization of $\mathbf{Z}$ but leading to a low value under another realization of $\mathbf{Z}$. In other words, the evaluation of $f$ at $\mathbf{x}$ is a random variable, denoted as $f(\mathbf{x}, \mathbf{Z})$, whose randomness is induced by $\mathbf{Z}$. Therefore, to determine the "optimal" design variable $\mathbf{x}_*$, one cannot directly compare the function evaluations at different values of $\mathbf{x}$. An alternative approach is to consider a risk measure of the distribution of $f(\mathbf{x}, \mathbf{Z})$ which accounts for the probability (or the risk) that $f(\mathbf{x}, \mathbf{Z})$ suffers from poor realizations. In this paper, we examine two risk measures for $f(\mathbf{x}, \mathbf{Z})$: *value-at-risk* (VaR), denoted as $v_f(\mathbf{x}; \alpha)$, and *conditional VaR* (CVaR), denoted as $c_f(\mathbf{x}; \alpha)$. Here, $\alpha \in (0, 1)$ represents a known *risk level*. VaR and CVaR are defined as follows:

$$v_f(\mathbf{x}; \alpha) \triangleq \inf\{\gamma :\ P(f(\mathbf{x}, \mathbf{Z}) \leq \gamma) \geq \alpha\} \qquad c_f(\mathbf{x}; \alpha) \triangleq \frac{1}{\alpha} \int_0^\alpha v_f(\mathbf{x}; \alpha')\, \mathrm{d}\alpha' . \qquad (1)$$

While $v_f(\mathbf{x}; \alpha)$ can be viewed as $100\alpha\%$-percentile of $f(\mathbf{x}, \mathbf{Z})$ (i.e., the event that $f(\mathbf{x}, \mathbf{Z})$ suffers from realizations lower than $v_f(\mathbf{x}; \alpha)$ happens with a probability of at most $\alpha$), $c_f(\mathbf{x}; \alpha)$ is the expected value of $f(\mathbf{x}, \mathbf{Z})$ in the worst $100\alpha\%$ of cases. To avoid unnecessary repetition, we unify the notations of both VaR and CVaR as $\rho_f(\mathbf{x}; \alpha)$. The notations $v_f(\mathbf{x}; \alpha)$ and $c_f(\mathbf{x}; \alpha)$ are only used when VaR and CVaR require distinct treatments.

Given the risk measure $\rho_f(\mathbf{x}; \alpha)$ and the distribution of $\mathbf{Z}$, the *BO of risk measure* problem aims to search for the maximizer $\mathbf{x}_* \triangleq \arg\max_{\mathbf{x} \in \mathcal{X}} \rho_f(\mathbf{x}; \alpha)$ (Cakmak et al., 2020). Since $f$ is unknown, determining this maximizer typically involves iteratively observing noisy evaluations $y(\mathbf{x}_t, \mathbf{z}_t) \triangleq f(\mathbf{x}_t, \mathbf{z}_t) + \epsilon(\mathbf{x}_t, \mathbf{z}_t)$ for iteration $t \geq 1$ where $\epsilon(\mathbf{x}_t, \mathbf{z}_t) \sim \mathcal{N}(0, \sigma_n^2)$ are i.i.d. Gaussian noises.[2] Furthermore, it is often expensive to obtain these evaluations (e.g., requiring running expensive simulations). Thus, the crux of the problem lies in devising a selection strategy of the *input query* $(\mathbf{x}_t, \mathbf{z}_t)$ at iteration $t$, considering observations in the previous $t - 1$ iterations, denoted as $\mathbf{y}_{\mathbf{D}_t} \triangleq (y(\mathbf{x}_{t'}, \mathbf{z}_{t'}))_{t'=1}^{t-1}$ where $\mathbf{D}_t \triangleq ((\mathbf{x}_{t'}, \mathbf{z}_{t'}))_{t'=1}^{t-1}$, to quickly discover $\mathbf{x}_*$.

We model $f$ with a *Gaussian process* (GP) (Rasmussen and Williams, 2006). Specifically, we use a constant GP prior mean function $\mu_0$ and a squared exponential kernel $k_{(\mathbf{x},\mathbf{z}),(\mathbf{x}',\mathbf{z}')} \triangleq \mathrm{cov}[f(\mathbf{x}, \mathbf{z}), f(\mathbf{x}', \mathbf{z}')] = \sigma_s^2 \exp(-0.5(\mathbf{x}-\mathbf{x}')^\top \Lambda_x^{-2}(\mathbf{x}-\mathbf{x}') - 0.5(\mathbf{z}-\mathbf{z}')^\top \Lambda_z^{-2}(\mathbf{z}-\mathbf{z}'))$ where $\Lambda_x \triangleq$

---

[2]We adopt the settings outlined in previous works of Cakmak et al. (2020); Nguyen et al. (2021a) such that $\mathbf{z}$ is controllable during the optimization (discussed in Sec. 1).

diag$(l_1, \ldots, l_d)$, $\Lambda_z \triangleq$ diag$(l_{d+1}, \ldots, l_{d+d'})$, and $\sigma_s^2$ are the lengthscales and the signal variance, respectively. Given $\mathbf{y}_{\mathbf{D}_t}$, the posterior belief of $f(\mathbf{x}, \mathbf{z})$ is a Gaussian distribution with the following mean and variance

$$\mu_t(\mathbf{x}, \mathbf{z}) \triangleq \mu_0 + \mathbf{K}_{(\mathbf{x},\mathbf{z}),\mathbf{D}_t} \Lambda_{\mathbf{D}_t} (\mathbf{y}_{\mathbf{D}_t} - \mu_0) \qquad \sigma_t^2(\mathbf{x}, \mathbf{z}) \triangleq k_{(\mathbf{x},\mathbf{z})} - \mathbf{K}_{(\mathbf{x},\mathbf{z}),\mathbf{D}_t} \Lambda_{\mathbf{D}_t} \mathbf{K}_{\mathbf{D}_t,(\mathbf{x},\mathbf{z})}$$

where $\Lambda_{\mathbf{D}_t} \triangleq \left( \mathbf{K}_{\mathbf{D}_t,\mathbf{D}_t} + \sigma_n^2 \mathbf{I} \right)^{-1}$, $\mathbf{K}$ is the covariance matrix, and $\mathbf{I}$ is the identity matrix.

The performance of a BO algorithm is often analyzed through the *cumulative regret*, denoted as $R_T \triangleq \sum_{t=1}^{T} r(\mathbf{x}_t)$ where $r(\mathbf{x}_t) \triangleq \rho_f(\mathbf{x}_*; \alpha) - \rho_f(\mathbf{x}_t; \alpha)$ is the *instantaneous regret* at the input query $\mathbf{x}_t$. The algorithm is *no-regret* if its cumulative regret is sublinear, i.e., $\lim_{T \to \infty} R_T/T \to 0$. In such a case, the *simple regret* $S_T \triangleq \min_{t=1,\ldots,T} r(\mathbf{x}_t) \leq R_T/T$ approaches 0. This suggests that one should recommend the best observed input so far as the outcome of the optimization, e.g., in the vanilla BO problem with a small noise. Unfortunately, in BO of risk measures, $\rho_f(\mathbf{x}_t; \alpha)$ is unobserved due to the unobserved $f(\mathbf{x}_t, \mathbf{z})$ for $(\mathbf{x}_t, \mathbf{z}) \notin \mathbf{D}_t$. This prevents us from determining the best observed input. Therefore, we will design a recommendation strategy such that the instantaneous regret $r(\mathbf{x}_t^\star)$ at the recommended input $\mathbf{x}_t^\star$ approaches zero as $t \to \infty$ in Sec. 3.3.

## 2.2 META-BO OF RISK MEASURES

Unlike existing works on BO of risk measures, our approach in this paper considers a set $\mathcal{T}$ of prior tasks. Let $f : \mathcal{X} \times \mathcal{Z} \to \mathbb{R}$ denote the blackbox function of the *current task* and $f_\tau : \mathcal{X} \times \mathcal{Z} \to \mathbb{R}$ denote the blackbox function in a prior task $\tau \in \mathcal{T}$. Furthermore, to facilitate later theoretical analysis, we follow the works of Chowdhury and Gopalan (2017) to assume that all blackbox functions belong to the *reproducing kernel Hilbert space* (RKHS) associated with the kernel $k$ and a bounded RKHS norm, i.e., $\|f\|_k \leq B$ and $\forall \tau \in \mathcal{T}$, $\|f_\tau\|_k \leq B$. The optimization result of a prior task $\tau \in \mathcal{T}$ includes the GP posterior mean and variance of $f_\tau$, denoted as $\hat{\mu}_\tau : \mathcal{X} \times \mathcal{Z} \to \mathbb{R}$ and $\hat{\sigma}_\tau^2 : \mathcal{X} \times \mathcal{Z} \to \mathbb{R}^+$, respectively. Meta-BO of the risk measure $\rho$ aims to utilize $(\hat{\mu}_\tau, \hat{\sigma}_\tau^2)_{\tau \in \mathcal{T}}$ to improve the optimization of $\rho_f$. However, the similarity between $f$ and $f_\tau$ (hence, between $\rho_f$ and $\rho_{f_\tau}$) is unknown for all $\tau \in \mathcal{T}$. In particular, if $\rho_{f_\tau}$ differs from $\rho_f$, naively utilizing $\hat{\mu}_\tau, \hat{\sigma}_\tau^2$ to optimize $\rho_f$ can deteriorate the optimization of $\rho_f$.

Under the assumption that blackbox functions belong to the RKHS space with a norm bounded by $B$, we first revisit the well-known confidence bound of $f(\mathbf{x}, \mathbf{z})$ from the work of Chowdhury and Gopalan (2017). Let $\gamma_t$ denote the maximum information gain about $f$ that can be obtained from any set of $t - 1$ observations. Choosing $\delta \in (0, 1)$, $\beta_t = (B + \sigma_n \sqrt{2(\gamma_t + 1 + \log 1/\delta)})^2$, then the event that $\forall t \geq 1, \forall \mathbf{x} \in \mathcal{X}, \forall \mathbf{z} \in \mathcal{Z}, l_t(\mathbf{x}, \mathbf{z}) \leq f(\mathbf{x}, \mathbf{z}) \leq u_t(\mathbf{x}, \mathbf{z})$ holds with probability $\geq 1 - \delta$ where the lower and upper confidence bounds are defined as

$$l_t(\mathbf{x}, \mathbf{z}) \triangleq \mu_t(\mathbf{x}, \mathbf{z}) - \beta_t^{1/2} \sigma_t(\mathbf{x}, \mathbf{z}) \quad \text{and} \quad u_t(\mathbf{x}, \mathbf{z}) \triangleq \mu_t(\mathbf{x}, \mathbf{z}) + \beta_t^{1/2} \sigma_t(\mathbf{x}, \mathbf{z}) \text{ , resp.} \quad (2)$$

Leveraging on the above confidence bounds $l_t$ and $u_t$, Nguyen et al. (2021b;a) propose the following confidence interval of the risk measure $\rho_f(\mathbf{x}; \alpha)$ (applicable to both VaR and CVaR): with probability $\geq 1 - \delta$,

$$\forall t \geq 1, \ \forall \mathbf{x} \in \mathcal{X}, \ \rho_f(\mathbf{x}; \alpha) \in [\rho_{l_t}(\mathbf{x}; \alpha), \rho_{u_t}(\mathbf{x}; \alpha)] \ . \quad (3)$$

where $l_t$ and $u_t$ are defined in equation (2) and $\rho_f, \rho_{l_t}, \rho_{u_t}$ can be replaced by $v_f, v_{l_t}$, and $v_{u_t}$ for VaR or by $c_f, c_{l_t}, c_{u_t}$ for CVaR in equation (1). For a prior task $\tau \in \mathcal{T}$, we denote the confidence interval of $f_\tau(\mathbf{x}, \mathbf{z})$ as $[\hat{l}_\tau(\mathbf{x}, \mathbf{z}), \hat{u}_\tau(\mathbf{x}, \mathbf{z})]$ which is defined in equation (2) using the GP posterior belief $(\hat{\mu}_\tau, \hat{\sigma}_\tau^2)$ of $f_\tau$ and the index of the last optimization iteration of task $\tau$. Then, the confidence interval of the risk measure $\rho_{f_\tau}(\mathbf{x}; \alpha)$ of prior task $\tau$ is $[\rho_{\hat{l}_\tau}(\mathbf{x}; \alpha), \rho_{\hat{u}_\tau}(\mathbf{x}; \alpha)]$.

## 3 OUR PROPOSED SOLUTION TO META-BO OF RISK MEASURES

Typically, at each iteration of the GP-UCB algorithm (Srinivas et al., 2010), there is only one option to select the input query from (assuming a unique maximizer of the GP-UCB acquisition function). Hence, in order to incorporate the information from prior tasks, the existing RM-GP-UCB (Dai et al., 2022) takes a weighted average of the GP-UCB acquisition functions of the current task and prior tasks. However, these weights depend on the maximum difference between $f$ and $f_\tau$ evaluations,

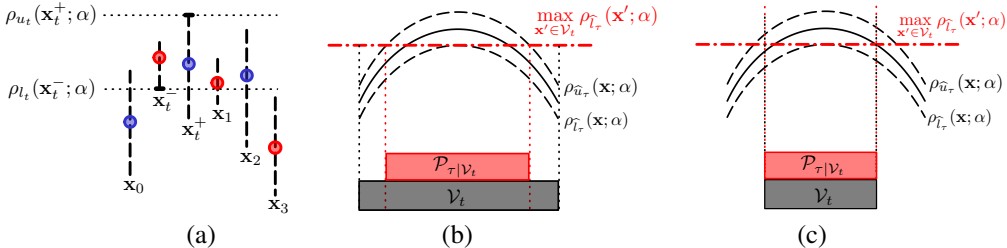

Figure 1: **(a)** Plot of $\mathcal{V}_t = \{\mathbf{x}_0, \mathbf{x}_t^+, \mathbf{x}_2\}$ with $\lambda = 0$ and $\eta = 1$. The dashed line shows the confidence interval of $\rho_f(\mathbf{x}_*; \alpha)$. While $\mathbf{x}_t^-$ and $\mathbf{x}_1$ violate UC, $\mathbf{x}_3$ violates OC. **(b)** Plot of an informative prior task $\tau$ where $\mathcal{V}_t \setminus \mathcal{P}_{\tau|\mathcal{V}_t} \neq \emptyset$. **(c)** Plot of an uninformative prior task $\tau$ where $\mathcal{P}_{\tau|\mathcal{V}_t} = \mathcal{V}_t$.

known as the *function gap*, at observed inputs. Defining such a gap notion for risk measures is not obvious. Moreover, the function gap is susceptible to function transformations such as scaling and vertical-shifting. Last, in order to achieve theoretical performance guarantee, RM-GP-UCB requires the weights of prior tasks to decay to zero.

In contrast, our meta-VBO approach tackles the problem from a different perspective that not only exhibits robustness against scaling and vertical-shifting transformations but also eliminates the explicit need for diminishing the effect of prior tasks via weight decay. Specifically, in the next section, we offer multiple choices of the input query in the form of a *versatile query set* (V-set), denoted as $\mathcal{V}_t$ (depending on iteration $t$), such that any input query sequence in the Cartesian product $\prod_{t=1}^T \mathcal{V}_t$ incurs a sublinear cumulative regret,

$$\forall (\mathbf{x}_t)_{t=1}^T \in \prod_{t=1}^T \mathcal{V}_t, \quad \lim_{T \to \infty} \frac{1}{T} R_T = \lim_{T \to \infty} \frac{1}{T} \sum_{t=1}^T r(\mathbf{x}_t) = 0. \tag{4}$$

The flexibility of choosing any input query $\mathbf{x}_t \in \mathcal{V}_t$ grants us the opportunity to incorporate information from prior tasks into the current optimization task as elaborated in Sec. 3.2.

## 3.1 VERSATILE QUERY SET

Intuitively, the *versatile query set* (V-set) $\mathcal{V}_t$ should **(C1)** *contain those inputs that are likely to be the maximizer of $\rho_f$*. Furthermore, we would like to avoid over-exploitation at inputs with well-estimated risk measures where gathering observations leads to a minimal progress towards discovering $\mathbf{x}_*$. This is done by introducing another criterion **(C2)** to *exclude inputs with well-estimated risk measures from $\mathcal{V}_t$*. These two key criteria (C1) and (C2) are formulated as follows.

**(C1)** To determine whether an input is likely to be the maximizer of $\rho_f$, we obtain the following confidence interval of $\rho_f(\mathbf{x}_*; \alpha)$ in App. A. With probability $\geq 1 - \delta$,

$$\forall t \geq 1, \ \rho_f(\mathbf{x}_*; \alpha) \in [\rho_{l_t}(\mathbf{x}_t^-; \alpha), \rho_{u_t}(\mathbf{x}_t^+; \alpha)], \tag{5}$$

where $\mathbf{x}_t^- \triangleq \arg\max_{\mathbf{x} \in \mathcal{X}} \rho_{l_t}(\mathbf{x}; \alpha)$ and $\mathbf{x}_t^+ \triangleq \arg\max_{\mathbf{x} \in \mathcal{X}} \rho_{u_t}(\mathbf{x}; \alpha)$. It is noted that $\mathbf{x}_t^+$ is the input query at iteration $t$ in V-UCB (Nguyen et al., 2021b) and CV-UCB (Nguyen et al., 2021a). For an input $\mathbf{x}'$ to be the maximizer $\mathbf{x}_*$ of $\rho_f$ (i.e., the above condition (C1)), the confidence interval of its risk measure $\rho_f(\mathbf{x}'; \alpha)$ should overlap with that of the maximum risk measure $\rho_f(\mathbf{x}_*; \alpha)$, i.e., $\rho_{u_t}(\mathbf{x}'; \alpha) \geq \rho_{l_t}(\mathbf{x}_t^-; \alpha)$ from equation (3) and equation (5). Hence, we propose to formulate (C1) as the following *overlapping condition* (OC): For an input $\mathbf{x}'$ to be in $\mathcal{V}_t$,

$$\rho_{u_t}(\mathbf{x}'; \alpha) \geq \rho_{l_t}(\mathbf{x}_t^-; \alpha) + \lambda \left( \rho_{u_t}(\mathbf{x}_t^+; \alpha) - \rho_{l_t}(\mathbf{x}_t^-; \alpha) \right) \quad \text{for } \lambda \in [0, 1]. \tag{6}$$

As $\lambda \to 1$, $\rho_{u_t}(\mathbf{x}'; \alpha) \to \rho_{u_t}(\mathbf{x}_t^+; \alpha)$, i.e., the overlap between the confidence interval of $\rho_f(\mathbf{x}'; \alpha)$ and that of $\rho_f(\mathbf{x}_*; \alpha)$ at iteration $t$ increases. When $\lambda = 1$, the set of inputs satisfying OC reduce to only the input query $\mathbf{x}_t^+$ of V-UCB (Nguyen et al., 2021b) or CV-UCB (Nguyen et al., 2021a).

**(C2)** We would like to exclude inputs with well-estimated risk measures by ensuring that the uncertainty of their risk measures is relatively high compared to that of the maximum risk measure $\rho_f(\mathbf{x}_*; \alpha)$. Specifically, we propose to formulate (C2) as the following *uncertainty condition*

(UC): For an input $\mathbf{x}'$ to be in $\mathcal{V}_t$,

$$\underbrace{\rho_{u_t}(\mathbf{x}';\alpha) - \rho_{l_t}(\mathbf{x}';\alpha)}_{\text{uncertainty of } \rho_f(\mathbf{x}';\alpha)} \geq \frac{1}{\eta}\Big( \underbrace{\rho_{u_t}(\mathbf{x}_t^+;\alpha) - \rho_{l_t}(\mathbf{x}_t^-;\alpha)}_{\text{uncertainty of } \rho_f(\mathbf{x}_*;\alpha)} \Big) \quad \text{for } \eta \in [1, 1/\lambda]. \tag{7}$$

If $\eta > 1/\lambda$, OC implies UC since $\rho_{l_t}(\mathbf{x}_t^-;\alpha) \geq \rho_{l_t}(\mathbf{x};\alpha) \ \forall \mathbf{x} \in \mathcal{X}$ (from the definition of $\mathbf{x}_t^-$). Hence, we are only interested in UC when $\eta \leq 1/\lambda$ as specified in equation (7).

To sum up, we define V-set $\mathcal{V}_t$ as the set of inputs $\mathbf{x} \in \mathcal{X}$ that satisfy both OC in equation (6) and UC in equation (7) (illustrated in Fig. 1a). The conditions of $\lambda \leq 1$ and $\eta \geq 1$ ensure $\mathcal{V}_t$ is always non-empty which follows from the following Lemma 3.1 (proved in App. B).

**Lemma 3.1.** *For all $t \geq 1$, $\mathcal{V}_t$ contains $\mathbf{x}_t^+$.*

We note that the size of $\mathcal{V}_t$ is non-decreasing when $\lambda$ decreases or $\eta$ increases. Specifically, $\lambda = 1$ implies $\eta = 1$ (since $\eta \in [1, 1/\lambda]$) and OC implies UC. In this case, $\mathcal{V}_t = \{\mathbf{x}_t^+\}$ (assuming $\rho_{u_t}$ has a unique maximizer), i.e., $\mathcal{V}_t$ reduces to the UCB-typed input query such as those in V-UCB (Nguyen et al., 2021b) and CV-UCB (Nguyen et al., 2021a). There is not any flexibility in choosing $\mathbf{x}_t$ from $\mathcal{V}_t = \{\mathbf{x}_t^+\}$, which prevents us from incorporating the information of prior tasks into the current task. However, when $\lambda < 1$ and $\eta \in [1, 1/\lambda]$, $\mathcal{V}_t$ contains other inputs than the popular input query $\mathbf{x}_t^+$ of UCB-typed algorithms. This is a trade-off between prior tasks (existing knowledge) and the current task (new knowledge). When $\lambda = 1$ ($\mathcal{V}_t = \{\mathbf{x}_t^+\}$), we utilize only observations from the current task. Conversely, decreasing $\lambda$ (increasing $|\mathcal{V}_t|$) promotes the exploitation/utilization of prior tasks to select an input query in $\mathcal{V}_t$. This *task-specific* trade-off is distinct from the conventional (*function-evaluation-specific*) exploration-exploitation trade-off associated with estimating the maximizer in V-UCB (Nguyen et al., 2021b) and CV-UCB (Nguyen et al., 2021a) (pertaining to a particular task).

Given the selected $\mathbf{x}_t \in \mathcal{V}_t$, we adopt the choice of $\mathbf{z}_t$ in the existing works of Nguyen et al. (2021b;a). In particular, if the risk measure is VaR, then $\mathbf{z}_t$ is selected as a *lacing value* (LV) w.r.t. $\mathbf{x}_t$, $l_t$, $u_t$, and $\alpha$.[3] If the risk measure is CVaR, we first find the risk level $\alpha_t \triangleq \arg\max_{\alpha' \in (0,\alpha]} v_{u_t}(\mathbf{x}_t;\alpha') - v_{l_t}(\mathbf{x}_t;\alpha')$. Then, $\mathbf{z}_t$ is selected as an LV w.r.t. $\mathbf{x}_t$, $l_t$, $u_t$, and $\alpha_t$.

Interestingly, given the above choice of $\mathbf{z}_t$, any choice of $\mathbf{x}_t$ in our proposed $\mathcal{V}_t$ results in a sublinear cumulative regret as shown in the following Theorem 3.2 (proved in App. D). In other words, we achieve the goal of constructing $\mathcal{V}_t$ that satisfies equation (4).

**Theorem 3.2.** *Given the selection of $\mathbf{x}_t$ as any input in $\mathcal{V}_t$ and the selection of $\mathbf{z}_t$ from the works of Nguyen et al. (2021b;a), the algorithm achieves a sublinear cumulative regret:*

$$\forall T \geq 1, R_T \leq (\eta(1-\lambda)+1)\sqrt{C_1 T \beta_T \gamma_T} \tag{8}$$

*holds with probability $\geq 1-\delta$ where $C_1 \triangleq 8/\log(1+\sigma_n^{-2})$; $\gamma_T$, $\beta_T$, $\delta$ are defined in equation (2); $\lambda$ and $\eta$ are defined in equation (6) and equation (7), respectively.*

Given the flexibility of choosing $\mathbf{x}_t$ in $\mathcal{V}_t$, we are now able to transfer the information from prior tasks to the current task by assigning different priorities to inputs in $\mathcal{V}_t$.

## 3.2 Assigning Priority to Inputs in Versatile Query Set $\mathcal{V}_t$

Since selecting any input query $\mathbf{x}_t \in \mathcal{V}_t$ results in a no-regret algorithm (Theorem 3.2), we would like to exploit the results of prior tasks to assign varying degrees of importance (i.e., *priorities*) to inputs in $\mathcal{V}_t$. To be robust to scaling and vertical-shifting of the blackbox function, the priority is based on the location of the maximizer rather than the estimated risk measure value. Besides, to provide more opportunities for transferring information between tasks, we consider the case that the global maximizer of the current task is the local maximizer of a prior task. Hence, we assign higher priorities to inputs that have a higher chance of being a local maximizer (restricted to $\mathcal{V}_t$) of prior tasks. For this reason, we propose the notion of *the probable local maximizer set*, denoted as $\mathcal{P}_{\tau|\mathcal{V}_t}$, which contains inputs that are probably a local maximizer restricted to $\mathcal{V}_t$ of $\rho_{f_\tau}$:

$$\mathcal{P}_{\tau|\mathcal{V}_t} \triangleq \{\mathbf{x} \in \mathcal{V}_t | \rho_{\widehat{u}_\tau}(\mathbf{x};\alpha) \geq \max_{\mathbf{x}' \in \mathcal{V}_t} \rho_{\widehat{l}_\tau}(\mathbf{x}';\alpha)\}. \tag{9}$$

---

[3]Lacing values $\mathbf{z}_{\text{LV}}$ w.r.t. $\mathbf{x}$, $l_t$, $u_t$, and $\alpha$ satisfy $l_t(\mathbf{x}, \mathbf{z}_{\text{LV}}) \leq v_{l_t}(\mathbf{x};\alpha) \leq v_{u_t}(\mathbf{x};\alpha) \leq u_t(\mathbf{x}, \mathbf{z}_{\text{LV}})$ (Nguyen et al., 2021b).

An alternative assumption is that the global maximizer of the current task is close to that of a prior task, then we can consider *the probable global maximizer set*, defined as $\mathcal{P}_{\tau|\mathcal{X}} \triangleq \{\mathbf{x} \in \mathcal{X}| \rho_{\widehat{u}_\tau}(\mathbf{x}; \alpha) \geq \max_{\mathbf{x}' \in \mathcal{X}} \rho_{\widehat{l}_\tau}(\mathbf{x}'; \alpha)\}$. Since using $\mathcal{P}_{\tau|\mathcal{X}}$ does not substantially change the implementation of meta-VBO in comparison to using $\mathcal{P}_{\tau|\mathcal{V}_t}$, we only focus on the probable local maximizer set $\mathcal{P}_{\tau|\mathcal{V}_t}$ in this paper. Figs. 1b and 1c show the set $\mathcal{P}_{\tau|\mathcal{V}_t}$ of inputs $\mathbf{x} \in \mathcal{V}_t$ such that $\rho_{\widehat{u}_\tau}(\mathbf{x}; \alpha)$ is above the dash-dotted line representing $\max_{\mathbf{x}' \in \mathcal{V}_t} \rho_{\widehat{l}_\tau}(\mathbf{x}'; \alpha)$). Essentially, we transfer the information about the *local maximizer* restricted to $\mathcal{V}_t$ of prior tasks to the current task. This is an unexplored direction in the current meta-BO literature.

Given the probable local maximizer set $\mathcal{P}_{\tau|\mathcal{V}_t}$, the priority of an input $\mathbf{x} \in \mathcal{V}_t$, denoted as $\pi_t(\mathbf{x})$, is measured by the number of the probable local maximizer sets that $\mathbf{x}$ belongs to, i.e.,

$$\pi_t(\mathbf{x}) \triangleq \sum_{\tau \in \mathcal{T}} \mathbb{1}_{\mathbf{x} \in \mathcal{P}_{\tau|\mathcal{V}_t}} . \tag{10}$$

The higher $\pi_t(\mathbf{x})$ is, the more likely that $\mathbf{x}$ is in the probable local maximizer set of a prior task. Hence, we select the input query $\mathbf{x}_t$ as

$$\mathbf{x}_t \triangleq \max_{\mathbf{x} \in \mathcal{V}_{t,\max}} \rho_{u_t}(\mathbf{x}; \alpha) \text{ where } \mathcal{V}_{t,\max} \triangleq \{\mathbf{x} \in \mathcal{V}_t| \pi_t(\mathbf{x}) = \max_{\mathbf{x}' \in \mathcal{V}_t} \pi_t(\mathbf{x}')\} . \tag{11}$$

*Remark* 3.3 (Prior harmful tasks). The local maximizers of $\rho_{f_\tau}$ in a prior harmful task restricted to $\mathcal{V}_t$ are different from that of $\rho_f$ of the current task. Contrary to the intuition suggesting that prior harmful tasks may bias meta-VBO away from the desirable maximizer $\mathbf{x}_*$, Theorem 3.2 asserts that our algorithm maintains a sublinear cumulative regret by constraining the choice of $\mathbf{x}_t$ to the $\mathcal{V}_t$. However, one could reasonably anticipate a potential reduction in the convergence rate, as empirically shown in Sec. 4, due to the influence of prior harmful tasks.

*Remark* 3.4 (Informativeness of a prior task). If $\pi_t(\mathbf{x})$ is constant for all $\mathbf{x} \in \mathcal{V}_t$, it results in $\mathcal{V}_{t,\max} = \mathcal{V}_t$ and $\mathbf{x}_t = \mathbf{x}_t^+$. This happens when there are not any prior tasks or when the BO results of prior tasks are *uninformative* in the local region $\mathcal{V}_t$, i.e., $\mathcal{P}_{\tau|\mathcal{V}_t} = \mathcal{V}_t \,\forall \tau \in \mathcal{T}$. As $\mathbf{x}_t = \mathbf{x}_t^+$ in these cases, our meta-VBO algorithm reduces to V-UCB (Nguyen et al., 2021b) or CV-UCB (Nguyen et al., 2021a). Fig. 1c illustrates an iteration $t$ where prior task $\tau$ is uninformative as $\mathcal{V}_t = \mathcal{P}_{\tau|\mathcal{V}_t}$, while Fig. 1b shows the case that $\tau$ is an informative task. It is noted that the informativeness of a prior task w.r.t. the current task depends on $\mathcal{V}_t$, hence, the iteration $t$. A prior task may be informative when we have not gathered many observations on the current task ($\mathcal{V}_t$ is large such that $\mathcal{P}_{\tau|\mathcal{V}_t}$ is a strict subset of $\mathcal{V}_t$), but it may become uninformative when many observations are acquired ($\mathcal{V}_t$ is small such that $\mathcal{P}_{\tau|\mathcal{V}_t} = \mathcal{V}_t$). When a prior task is uninformative, it does not influence the optimization of the current task in our meta-VBO algorithm (from the definition of $\pi_t(\mathbf{x})$ in equation (39)). Hence, the effect of a prior task on the current task is adaptively controlled by $\mathcal{V}_t$ as illustrated in Appendix I.1. This approach contrasts with that of Dai et al. (2022) that explicitly requires a schedule of the weight decay based on the function gap.

*Remark* 3.5 (Scaling and vertical-shifting of the blackbox functions). Our proposed meta-VBO depends on $\mathcal{V}_t$ and $\mathcal{P}_{\tau|\mathcal{V}_t}$. These sets rely on comparing risk measures at different inputs which remains unchanged under scaling or vertical-shifting of the blackbox function. Hence, our algorithm is robust to these transformations.

*Remark* 3.6 (Meta-VBO for vanilla BO and adversarially robust BO). As our proposed meta-VBO depends on the upper and lower confidence bounds of function evaluations, it is sufficiently general to address the vanilla meta-BO problem (see Appendix F). Its potential applicability to the meta-learning of other UCB-typed algorithms remains an area for future exploration. Furthermore, since adversarially robust BO is a special case of BO of VaR $\alpha \to 0^+$ (Nguyen et al., 2021b), our proposed meta-VBO directly applies to adversarially robust BO (see App. G).

## 3.3 DECISION VARIABLE RECOMMENDATION STRATEGY

At iteration $t$, we propose to recommend $\mathbf{x}_t^\star \triangleq \mathbf{x}_{\zeta(t)}^-$ where $\zeta(t) \triangleq \arg\max_{t' \in \{1,\ldots,t\}} \rho_{l_{t'}}(\mathbf{x}_{t'}^-; \alpha)$. We prove that its instantaneous regret approaches zero in App. E.

**Theorem 3.7.** *Let* $\zeta(t) \triangleq \arg\max_{t' \in \{1,\ldots,t\}} \rho_{l_{t'}}(\mathbf{x}_{t'}^-; \alpha)$, *then, by recommending* $\mathbf{x}_t^\star = \mathbf{x}_{\zeta(t)}^-$, *the instantaneous regret at* $\mathbf{x}_t^\star$ *is bounded:* $r(\mathbf{x}_t^\star) \leq \eta\sqrt{C_1\beta_t\gamma_t/t}$ *with probability* $\geq 1-\delta$ *where* $\eta$, $C_1$, $\beta_t$, *and* $\gamma_t$ *are defined in Theorem 3.2.*

As sublinear bounds of $\gamma_t$ are established for common kernels such as the SE kernel in the work of Srinivas et al. (2010), the instantaneous regret at $\mathbf{x}_t^\star \triangleq \mathbf{x}_{\zeta(t)}^-$ approaches zero as $t \to \infty$. Furthermore, we should choose $\eta = 1$ to minimize the upper confidence bound of $r(\mathbf{x}_t^\star)$ in Theorem 3.7. Given $\eta = 1$, we choose $\lambda \in [0, 1]$ to trade off between the flexibility in choosing $\mathbf{x}_t$ (i.e., the size of V-set) and the upper confidence bound of the cumulative regret $R_T$ at the input queries in Theorem 3.2. In the experiments, as we want to exploit the solutions of prior tasks, we maximize the size of V-set by setting $\lambda = 0$. In this case, the upper confidence bound of $R_T$ is two times that of V-UCB (Nguyen et al., 2021b) and CV-UCB (Nguyen et al., 2021a): $R_T \le 2\sqrt{C_1 T \beta_T \gamma_T}$.

## 4 EXPERIMENTS

Given the absence of prior research on meta-BO of risk measures, we compare the performance of our proposed algorithm under different sets of prior tasks to that of V-UCB (Nguyen et al., 2021b) which is equivalent to our proposed algorithm when there are not any prior tasks, i.e., $\mathcal{T} = \emptyset$. Recall that our emphasis in this paper is on transferring the optimization results, rather than transferring the surrogate model, as discussed in Sec. 1. Therefore, we exclusively focus on independent GPs. They are also more practical due to their lower computational overhead, while still enabling us to provide fair comparisons to empirically demonstrate our advantage over V-UCB.[4] Our goal is to empirically verify whether our proposed algorithm is (i) able to exploit the information from prior informative tasks to improve its converge rate in comparison to V-UCB even if harmful tasks exist in $\mathcal{T}$ and (ii) robust to prior harmful tasks even if there are not any useful tasks in $\mathcal{T}$. To achieve this, we design different sets of prior tasks: $\mathcal{T}_{\texttt{useful-pos-scale}}$ (i.e., positive scaling of the blackbox function of the current task: $f_\tau(\mathbf{x}, \mathbf{z}) = af(\mathbf{x}, \mathbf{z})$ for $a > 0$), $\mathcal{T}_{\texttt{useful-vshift}}$ (i.e., vertical-shifting $f_\tau(\mathbf{x}, \mathbf{z}) = f(\mathbf{x}, \mathbf{z}) + b$ for $b \in \mathbb{R}$), $\mathcal{T}_{\texttt{harmful-neg-scale}}$ (i.e., negative scaling $f_\tau(\mathbf{x}, \mathbf{z}) = -f(\mathbf{x}, \mathbf{z})$), and $\mathcal{T}_{\texttt{harmful-hshift}}$ (i.e., horizontal-shifting $f_\tau(\mathbf{x}, \mathbf{z}) = f(\mathbf{x} + \xi, \mathbf{z})$ for $\xi \in \mathbb{R}^d$). Additionally, $\mathcal{T}_{\texttt{all}}$ is the union of the above 4 sets. We note that $\mathcal{T}_{\texttt{harmful-neg-scale}}$ contains the most harmful task as the function evaluations are negated. On the other hand, horizontal-shifting along the $\mathbf{x}$ dimensions in $\mathcal{T}_{\texttt{harmful-hshift}}$ moves the location of the maximizers away from that of the current task. As explained in Sec. 3.3, we set $\lambda = 0$ and $\eta = 1$. We present additional experimental results of meta-BO for CVaR in App. I.

The experiments are performed on 6 synthetic functions: a Gaussian curve, the Branin-Hoo, the Goldstein-Price, the six-hump camel, the Hartmann-3D, and the Hartmann-6D (obtained from https://www.sfu.ca/~ssurjano); and using 2 real-world datasets: the yacht hydrodynamics dataset (Dua and Graff, 2017) and a portfolio optimization dataset (obtained from the existing works of Cakmak et al. (2020); Nguyen et al. (2021a)). In the first 4 experiments, both the dimensions of $\mathbf{x}$ and $\mathbf{z}$ are set to $d = d' = 1$. In the Hartmann-3D experiment, $d = 2$ and $d' = 1$, while in the Hartmann-6D experiment, $d = 5$ and $d' = 1$. In the yacht hydrodynamics experiment, the optimization problem is to minimize VaR of the residuary resistance per unit weight of displacement of a yacht by controlling the its 5-dimensional hull geometry coefficients. The environmental random variable is the 1-dimensional Froude number. The blackbox function is generated by fitting a GP to the yacht hydrodynamics dataset (Dua and Graff, 2017). In the portfolio optimization experiment, we follow the work of (Cakmak et al., 2020) to optimize a portfolio by controlling $d = 3$ decision variables subject to the randomness in the bid-ask spread and the borrow cost (i.e., $d' = 2$). The blackbox function is generated by fitting a GP to the simulated portfolio dataset. The risk level is set at $\alpha = 0.1$ for all experiments.

The average and the standard error of $r(\mathbf{x}_t^\star)$ (in Sec. 3.3) over 30 random repetitions are shown in Fig. 2. It is noted that the curves of useful-pos-scale, useful-vshift, and all overlap each other in Figs. 2a, 2e, and 2g. In these figures, we observe that when $\mathcal{T}$ contains only prior useful tasks in $\mathcal{T}_{\texttt{useful-pos-scale}}$ and $\mathcal{T}_{\texttt{useful-vshift}}$, our proposed meta-VBO outperforms V-UCB which does not utilize any prior tasks. Notably, even when $\mathcal{T}$ contains both harmful and useful tasks in $\mathcal{T}_{\texttt{all}}$, meta-VBO still outperforms V-UCB. In the extreme case when $\mathcal{T}$ contains only harmful tasks in $\mathcal{T}_{\texttt{harmful-neg-scale}}$ and $\mathcal{T}_{\texttt{harmful-hshift}}$, meta-VBO does not outperform V-UCB, but it still converges to a low-regret solution. In Fig. 2f, meta-VBO given the set $\mathcal{T}_{\texttt{harmful-neg-scale}}$ performs poorly probably due to the large input space and the function surface of the Hartmann-6D. We can

---

[4]When computation cost are affordable, one may combine a method that transfers the surrogate model with our algorithm by simply replacing independent GPs with a multitask GP because our algorithm only relies on upper and lower confidence bounds that are readily obtainable from a multitask GP model.

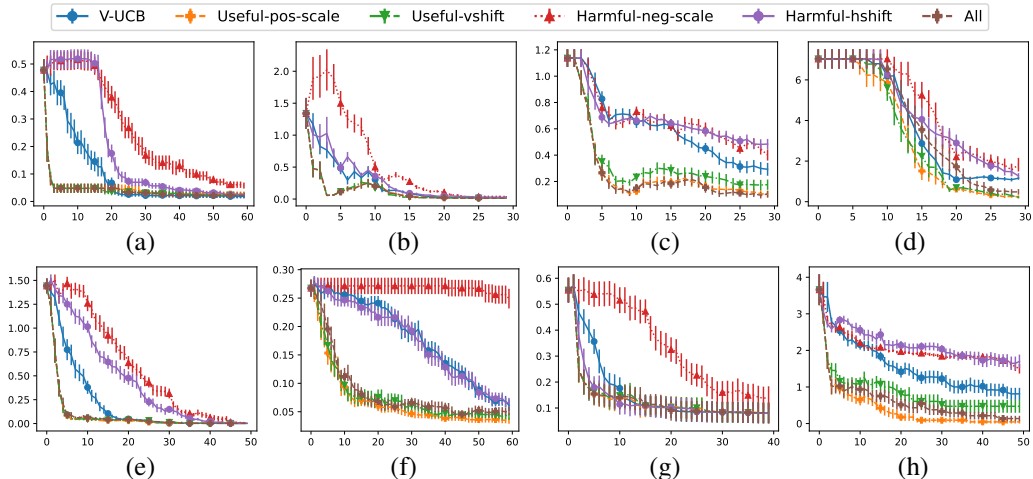

Figure 2: Plots of the regret at the recommended input against the BO iteration in experiments with (a) a Gaussian curve, (b) the Branin-Hoo function, (c) the Goldstein-Price function, (d) the six-hump camel function, (e) the Hartmann-3D function, (f) the Hartmann-6D function, (g) the yacht hydrodynamics dataset, and (h) the portfolio optimization dataset.



Figure 3: Percentage of prior tasks whose probable local maximizer sets contain the input query (grouped by the transformation) at different BO iterations in the experiment with the Gaussian curve and the set $\mathcal{T}_{all}$ of prior tasks. The percentage is computed over 30 repeated experiments.

also observe the percentage of prior tasks whose probable maximizer sets $\mathcal{P}_{\tau|\mathcal{V}_t}$ containing $\mathbf{x}_t$ in Fig. 3. It shows that as we gather more observations, the percentage of harmful tasks whose $\mathcal{P}_{\tau|\mathcal{V}_t}$ contains $\mathbf{x}_t$ reduces.

## 5  CONCLUSION

In this paper, we propose meta-VBO which is the first algorithm for meta-BO of risk measures by proposing a new concept of V-set and leveraging prior tasks to prioritize inputs in this set. Our algorithm offers a theoretical performance guarantee even when prior tasks are harmful. Furthermore, we propose a recommendation strategy that suggests an input as the result of the optimization, with a proof of its instantaneous regret approaching zero. The empirical performance of the proposed solution is demonstrated through several synthetic function benchmarks and real-world objective functions. In the future, we would like to study our meta-VBO approach to perform meta-learning for other BO variants such as federated BO (Dai et al., 2020), preferential BO (González et al., 2017; Nguyen et al., 2021c), BO for finding Nash equilibria (Tay et al., 2023), and BO with unknown constraints (Nguyen et al., 2023).

## REPRODUCIBILITY STATEMENT

To ensure the reproducibility of both the theoretical and experimental results in our paper, we have clearly elaborated on our assumptions, the proofs of the theoretical results, and provided the implementation of the experiments. In particular, our main assumptions are stated in the third paragraph in Sec. 1 and the first paragraph in Sec. 3. The details proofs of our theoretical results are shown in Appendices A, B, D, E, and G.

ACKNOWLEDGEMENT

This research is supported by AI Singapore, under grant AISG2-RP-2020-018.

This research is supported by the National Research Foundation (NRF), Prime Minister's Office, Singapore under its Campus for Research Excellence and Technological Enterprise (CREATE) programme. The Mens, Manus, and Machina (M3S) is an interdisciplinary research group (IRG) of the Singapore MIT Alliance for Research and Technology (SMART) centre.

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
