holds with probability $\geq 1 - \delta$.

*Proof.* Let us assume that $\rho_f(\mathbf{x}; \alpha)$ is in its confidence interval, i.e., the event $\forall t \geq 1, \forall \mathbf{x} \in \mathcal{X}, \rho_f(\mathbf{x}; \alpha) \in [\rho_{l_t}(\mathbf{x}; \alpha); \rho_{u_t}(\mathbf{x}; \alpha)]$ holds (which happens with probability $\geq 1 - \delta$). Furthermore, we define the following inputs in equation (5).

$$\mathbf{x}_t^- \triangleq \arg\max_{\mathbf{x} \in \mathcal{X}} \rho_{l_t}(\mathbf{x}; \alpha) \qquad \mathbf{x}_t^+ \triangleq \arg\max_{\mathbf{x} \in \mathcal{X}} \rho_{u_t}(\mathbf{x}; \alpha) .$$

Then, $\forall \mathbf{x} \in \mathcal{X}, \ \forall t \geq 1, \ \rho_f(\mathbf{x}_*; \alpha) \geq \rho_f(\mathbf{x}; \alpha) \geq \rho_{l_t}(\mathbf{x}; \alpha)$. Hence,

$$\forall t \geq 1, \ \rho_f(\mathbf{x}_*; \alpha) \geq \max_{\mathbf{x} \in \mathcal{X}} \rho_{l_t}(\mathbf{x}; \alpha)$$

$$\text{i.e.,} \ \forall t \geq 1, \ \rho_f(\mathbf{x}_*; \alpha) \geq \rho_{l_t}(\mathbf{x}_t^-; \alpha) . \tag{12}$$

Furthermore,

$$\forall t \geq 1, \ \rho_f(\mathbf{x}_*; \alpha) \leq \rho_{u_t}(\mathbf{x}_*; \alpha) \leq \max_{\mathbf{x} \in \mathcal{X}} \rho_{u_t}(\mathbf{x}; \alpha) = \rho_{u_t}(\mathbf{x}_t^+; \alpha) . \tag{13}$$

From equation (12) and equation (13),

$$\forall t \geq 1, \ \rho_f(\mathbf{x}_*; \alpha) \in [\rho_{l_t}(\mathbf{x}_t^-; \alpha), \rho_{u_t}(\mathbf{x}_t^+; \alpha)] .$$

$\square$

## B PROOF OF LEMMA 3.1

In this section, we prove that when $\lambda \leq 1$ and $\eta \geq 1$, $\mathbf{x}_t^+ \in \mathcal{V}_t$.

*Proof.* To show that $\mathbf{x}_t^+ \in \mathcal{V}_t$, we show that $\mathbf{x}_t^+$ satisfies both OC equation (6) and UC equation (7) conditions.

- **OC:** We observe that when $\lambda \leq 1$,

$$\rho_{u_t}(\mathbf{x}_t^+; \alpha) = \rho_{l_t}(\mathbf{x}_t^-; \alpha) + \left( \rho_{u_t}(\mathbf{x}_t^+; \alpha) - \rho_{l_t}(\mathbf{x}_t^-; \alpha) \right)$$
$$\geq \rho_{l_t}(\mathbf{x}_t^-; \alpha) + \lambda \left( \rho_{u_t}(\mathbf{x}_t^+; \alpha) - \rho_{l_t}(\mathbf{x}_t^-; \alpha) \right) .$$

  Hence, $\mathbf{x}_t^+$ satisfies OC equation (6) when $\lambda \leq 1$.

- **UC:** We observe that when $\eta \geq 1$,

$$\rho_{u_t}(\mathbf{x}_t^+; \alpha) - \rho_{l_t}(\mathbf{x}_t^+; \alpha) \geq \rho_{u_t}(\mathbf{x}_t^+; \alpha) - \rho_{l_t}(\mathbf{x}_t^-; \alpha) \geq \frac{1}{\eta} \left( \rho_{u_t}(\mathbf{x}_t^+; \alpha) - \rho_{l_t}(\mathbf{x}_t^-; \alpha) \right) .$$

  Hence, $\mathbf{x}_t^+$ satisfies UC equation (7) when $\eta \geq 1$.

$\square$

## C REVIEW OF THE UPPER CONFIDENCE BOUND OF THE $\rho_{u_t}(\mathbf{x}_t; \alpha) - \rho_{l_t}(\mathbf{x}_t; \alpha)$

Recall that the *lacing value* (LV), denoted as $\mathbf{z}_{\text{LV}}$ w.r.t. $\mathbf{x}$, $l_t$, $u_t$, and $\alpha$, is defined as the input in $\mathcal{Z}$ that satisfies the following property (Nguyen et al., 2021b)

$$l_t(\mathbf{x}, \mathbf{z}_{\text{LV}}) \leq v_{l_t}(\mathbf{x}; \alpha) \leq v_{u_t}(\mathbf{x}; \alpha) \leq u_t(\mathbf{x}, \mathbf{z}_{\text{LV}}) \tag{14}$$

The existence of LV is proved in (Nguyen et al., 2021b). It plays an important role in the choice of $\mathbf{z}_t$ given $\mathbf{x}_t$ in V-UCB (Nguyen et al., 2021b) and CV-UCB (Nguyen et al., 2021a). Specifically, given the input query $\mathbf{x}_t$, the works of Nguyen et al. (2021b;a) select $\mathbf{z}_t$ as follows:

- If we are optimizing VaR of $f(\mathbf{x}, \mathbf{Z})$, $\mathbf{z}_t$ is selected as an LV w.r.t. $\mathbf{x}_t$, $l_t$, $u_t$, and $\alpha$ (Nguyen et al., 2021b).
- If we are optimizing CVaR of $f(\mathbf{x}, \mathbf{Z})$, $\mathbf{z}_t$ is selected as an LV w.r.t. $\mathbf{x}_t$, $l_t$, $u_t$, and $\alpha_t \triangleq \arg\max_{\alpha' \in (0, \alpha]} v_{u_t}(\mathbf{x}_t; \alpha') - v_{l_t}(\mathbf{x}_t; \alpha')$ (Nguyen et al., 2021a).

We repeat the derivations in the works of Nguyen et al. (2021b) and Nguyen et al. (2021a) to show that with probability $\geq 1 - \delta$, for any input query $\mathbf{x}_t \in \mathcal{X}$ and the above choice of $\mathbf{z}_t$,

$$\rho_{u_t}(\mathbf{x}_t; \alpha) - \rho_{l_t}(\mathbf{x}_t; \alpha) \leq 2\beta_t^{1/2}\sigma_t(\mathbf{x}_t, \mathbf{z}_t) . \tag{15}$$

It is noted that equation (15) does not require any specific choice of $\mathbf{x}_t$. Due to the different choices of $\mathbf{z}_t$ in the optimization of VaR and CVaR, they are treated separately as follows.

## C.1 OPTIMIZATION OF VALUE-AT-RISK

As $\mathbf{z}_t$ is selected as an LV w.r.t. $\mathbf{x}_t$, $l_t$, $u_t$, and $\alpha$, it follows from the definition of LV equation (14) that

$$v_{u_t}(\mathbf{x}_t; \alpha) - v_{l_t}(\mathbf{x}_t; \alpha) \leq u_t(\mathbf{x}_t, \mathbf{z}_t) - l_t(\mathbf{x}_t, \mathbf{z}_t) = 2\beta_t^{1/2}\sigma_t(\mathbf{x}_t, \mathbf{z}_t) . \tag{16}$$

## C.2 OPTIMIZATION OF CONDITIONAL VALUE-AT-RISK

From the definition of CVaR in Sec. 2.1.

$$c_{u_t}(\mathbf{x}_t; \alpha) - c_{l_t}(\mathbf{x}_t; \alpha) = \frac{1}{\alpha} \int_0^\alpha v_{u_t}(\mathbf{x}_t; \alpha') - v_{l_t}(\mathbf{x}_t; \alpha') \, \mathrm{d}\alpha' .$$

Recall that $\alpha_t \triangleq \arg\max_{\alpha' \in (0, \alpha]} v_{u_t}(\mathbf{x}_t; \alpha') - v_{l_t}(\mathbf{x}_t; \alpha')$,

$$\forall \alpha' \in (0, \alpha], \ v_{u_t}(\mathbf{x}_t; \alpha') - v_{l_t}(\mathbf{x}_t; \alpha') \leq v_{u_t}(\mathbf{x}_t; \alpha_t) - v_{l_t}(\mathbf{x}_t; \alpha_t)$$

$$\text{i.e., } \frac{1}{\alpha} \int_0^\alpha v_{u_t}(\mathbf{x}_t; \alpha') - v_{l_t}(\mathbf{x}_t; \alpha') \, \mathrm{d}\alpha' \leq \frac{1}{\alpha} \int_0^\alpha v_{u_t}(\mathbf{x}_t; \alpha_t) - v_{l_t}(\mathbf{x}_t; \alpha_t) \, \mathrm{d}\alpha'$$

$$= v_{u_t}(\mathbf{x}_t; \alpha_t) - v_{l_t}(\mathbf{x}_t; \alpha_t)$$

Therefore,

$$c_{u_t}(\mathbf{x}_t; \alpha) - c_{l_t}(\mathbf{x}_t; \alpha) \leq v_{u_t}(\mathbf{x}_t; \alpha_t) - v_{l_t}(\mathbf{x}_t; \alpha_t)$$

Since $\mathbf{z}_t$ is selected as an LV w.r.t. $\mathbf{x}_t$, $l_t$, $u_t$, and $\alpha_t$,

$$v_{u_t}(\mathbf{x}_t; \alpha_t) - v_{l_t}(\mathbf{x}_t; \alpha_t) \leq u_t(\mathbf{x}_t, \mathbf{z}_t) - l_t(\mathbf{x}_t, \mathbf{z}_t) .$$

Hence,

$$c_{u_t}(\mathbf{x}_t; \alpha) - c_{l_t}(\mathbf{x}_t; \alpha) \leq u_t(\mathbf{x}_t, \mathbf{z}_t) - l_t(\mathbf{x}_t, \mathbf{z}_t)$$

$$= 2\beta_t^{1/2}\sigma_t(\mathbf{x}_t, \mathbf{z}_t) . \tag{17}$$

As a result, equation (15) follows from equation (16) and equation (17).

## D   PROOF OF THEOREM 3.2

First, we prove an upper confidence bound of the instantaneous regret $r(\mathbf{x}_t) \triangleq \rho_f(\mathbf{x}_*; \alpha) - \rho_f(\mathbf{x}_t; \alpha)$ in the following lemma.

**Lemma D.1.** *Given $\mathbf{x}_t \in \mathcal{V}_t$, i.e., $\mathbf{x}_t$ satisfies both OC equation (6) and UC equation (7), and $\mathbf{z}_t$ is selected as an LV w.r.t. $\mathbf{x}_t$, $l_t$, $u_t$, and $\alpha$ (if optimizing VaR) or $\alpha_t \triangleq \arg\max_{\alpha' \in (0, \alpha]} v_{u_t}(\mathbf{x}_t; \alpha') - v_{l_t}(\mathbf{x}_t; \alpha')$ (if optimizing CVaR), then*

$$\forall t \geq 1, \ r(\mathbf{x}_t) \leq 2\left(\eta(1 - \lambda) + 1\right)\beta_t^{1/2}\sigma_t(\mathbf{x}_t, \mathbf{z}_t)$$

*holds with probability $\geq 1 - \delta$.*

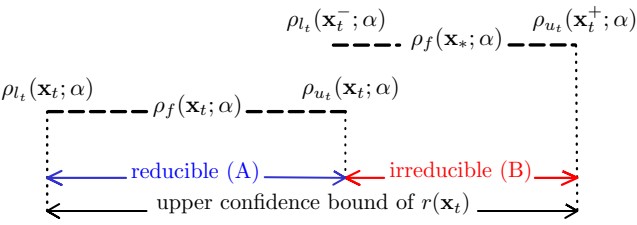

Figure 4: Decomposition an upper confidence bound of $r(\mathbf{x}_t)$.

*Proof.* Assuming that $\forall \mathbf{x} \in \mathcal{X}, \ \forall t \geq 1, \ \rho_f(\mathbf{x}; \alpha) \in [\rho_{l_t}(\mathbf{x}; \alpha), \rho_{u_t}(\mathbf{x}; \alpha)]$ which happens with probability $\geq 1 - \delta$.

We decompose an upper confidence bound of the instantaneous regret $r(\mathbf{x}_t)$ into two parts.

$$
\begin{aligned}
r(\mathbf{x}_t) &\triangleq \rho_f(\mathbf{x}_*; \alpha) - \rho_f(\mathbf{x}_t) \\
&= \rho_{u_t}(\mathbf{x}_*) - \rho_{l_t}(\mathbf{x}_t; \alpha) \\
&\leq \rho_{u_t}(\mathbf{x}_t^+; \alpha) - \rho_{l_t}(\mathbf{x}_t; \alpha) \\
&= \underbrace{\left[ \rho_{u_t}(\mathbf{x}_t^+; \alpha) - \rho_{u_t}(\mathbf{x}_t; \alpha) \right]}_{\text{irreducible (B)}} + \underbrace{\left[ \rho_{u_t}(\mathbf{x}_t; \alpha) - \rho_{l_t}(\mathbf{x}_t; \alpha) \right]}_{\text{reducible (A)}} .
\end{aligned}
\tag{18}
$$

Hence, an upper confidence bound $\rho_{u_t}(\mathbf{x}_t^+; \alpha) - \rho_{l_t}(\mathbf{x}_t; \alpha)$ of $r(\mathbf{x}_t)$ is decomposed into (A) a *reducible* part and (B) an *irreducible* part. (A) is called the reducible part because gathering observations at $\mathbf{x}_t$ reduces the uncertainty of $\rho_f(\mathbf{x}_t; \alpha)$, hence, reducing (A), while (B) is called the irreducible part because it is likely that gathering observations at $\mathbf{x}_t$ does not reduce the (B) (as $\rho_f(\mathbf{x}_t; \alpha) \leq \rho_{u_t}(\mathbf{x}_t; \alpha)$ with high probability). As a result, the primary difficulty is in bounding this irreducible part (A).

Fortunately, we select $\mathbf{x}_t \in \mathcal{V}_t$ such that it satisfies both (OC) equation (6) and (UC) equation (7) conditions, so the irreducible part (B) can be associated with the reducible part (A) as follows.

$$
\begin{aligned}
&\underbrace{\rho_{u_t}(\mathbf{x}_t^+; \alpha) - \rho_{u_t}(\mathbf{x}_t; \alpha)}_{\text{irreducible (B)}} \\
&= \left[ \rho_{u_t}(\mathbf{x}_t^+; \alpha) - \rho_{l_t}(\mathbf{x}_t^-; \alpha) \right] - \left[ \rho_{u_t}(\mathbf{x}_t; \alpha) - \rho_{l_t}(\mathbf{x}_t^-; \alpha) \right] \\
&\leq \left[ \rho_{u_t}(\mathbf{x}_t^+; \alpha) - \rho_{l_t}(\mathbf{x}_t^-; \alpha) \right] - \lambda \left[ \rho_{u_t}(\mathbf{x}_t^+; \alpha) - \rho_{l_t}(\mathbf{x}_t^-; \alpha) \right] \qquad \text{from (OC) equation (6)} \\
&= (1 - \lambda) \left[ \rho_{u_t}(\mathbf{x}_t^+; \alpha) - \rho_{l_t}(\mathbf{x}_t^-; \alpha) \right] \\
&\leq \eta(1 - \lambda) \Big[ \underbrace{\rho_{u_t}(\mathbf{x}_t; \alpha) - \rho_{l_t}(\mathbf{x}_t; \alpha)}_{\text{reducible (A)}} \Big] \qquad \text{from (UC) equation (7) and } \lambda \leq 1 .
\end{aligned}
\tag{19}
$$

Plug equation (19) into equation (18),

$$
\begin{aligned}
r(\mathbf{x}_t) &\leq \eta(1 - \lambda) \left[ \rho_{u_t}(\mathbf{x}_t; \alpha) - \rho_{l_t}(\mathbf{x}_t; \alpha) \right] + \left[ \rho_{u_t}(\mathbf{x}_t; \alpha) - \rho_{l_t}(\mathbf{x}_t; \alpha) \right] \\
&= (\eta(1 - \lambda) + 1) \left[ \rho_{u_t}(\mathbf{x}_t; \alpha) - \rho_{l_t}(\mathbf{x}_t; \alpha) \right] \\
&\leq 2 \left( \eta(1 - \lambda) + 1 \right) \beta_t^{1/2} \sigma_t(\mathbf{x}_t, \mathbf{z}_t) \quad \text{from equation (15)} .
\end{aligned}
$$

$\square$

*Remark* D.2. The reader may observe that one intention behind of (OC) and (UC) is to ensure the relationship between the irreducible part (B) and the reducible part (A) in equation (19). Then, instead of imposing 2 conditions (OC) and (UC), he/she may suggest we directly define an alternative versatile query set $\widetilde{\mathcal{Q}}_t$ by imposing this relationship, e.g.,

$$
\widetilde{\mathcal{Q}}_t \triangleq \{ \mathbf{x} \in \mathcal{X} | \ \rho_{u_t}(\mathbf{x}_t^+; \alpha) - \rho_{u_t}(\mathbf{x}; \alpha) \leq \nu(\rho_{u_t}(\mathbf{x}; \alpha) - \rho_{l_t}(\mathbf{x}; \alpha)) \}
\tag{20}
$$

for $\nu \geq 0$. However, the above definition equation (20) of $\widetilde{\mathcal{Q}}_t$ is insufficient for the goal of optimizing $\rho_f(\mathbf{x}; \alpha)$. This is because $\widetilde{\mathcal{Q}}_t$ contains those inputs $\mathbf{x} \in \mathcal{X}$ that do not satisfies (OC) equation (6),

i.e., $\rho_{u_t}(\mathbf{x}; \alpha) < \rho_{l_t}(\mathbf{x}_t^-; \alpha)$, which are unlikely to be the maximizer of $\rho_f(\mathbf{x}; \alpha)$. Hence, sampling at these inputs is inefficient as they may not provide any information on the maximizer $\mathbf{x}_*$. Furthermore, while (OC) and (UC) are interpretable as described in Sec. 3.1, interpreting the definition of $\widetilde{\mathcal{Q}}_t$ in equation (20) is more challenging. As a result, the versatile query set $\mathcal{V}_t$ is defined based on (OC) and (UC) conditions which naturally lead to the relationship in equation (19).

Given Lemma D.1, with probability $\geq 1 - \delta$, the cumulative regret $R_T$ is bounded by

$$
\begin{aligned}
R_T &\triangleq \sum_{t=1}^{T} r(\mathbf{x}_t) \\
&= 2(\eta(1 - \lambda) + 1) \sum_{t=1}^{T} \beta_t^{1/2} \sigma_t(\mathbf{x}_t, \mathbf{z}_t) \qquad \text{(from Lemma D.1)} \\
&\leq 2(\eta(1 - \lambda) + 1) \beta_T^{1/2} \sum_{t=1}^{T} \sigma_t(\mathbf{x}_t, \mathbf{z}_t) \qquad \text{(from the monotonicity of } \beta_t\text{)} .
\end{aligned}
$$

To prove Theorem 3.2, it remains to show that

$$
2\beta_T^{1/2} \sum_{t=1}^{T} \sigma_t(\mathbf{x}_t, \mathbf{z}_t) \leq \sqrt{C_1 T \beta_T \gamma_T} \tag{21}
$$

where $C_1 \triangleq 8/\log(1 + \sigma_n^{-2})$, which follows directly from the work of Srinivas et al. (2010). Specifically, using Cauchy-Schwarz inequality,

$$
4\beta_T \left( \sum_{t=1}^{T} \sigma_t(\mathbf{x}_t, \mathbf{z}_t) \right)^2 \leq 4\beta_T T \sum_{t=1}^{T} \sigma_t^2(\mathbf{x}_t, \mathbf{z}_t) .
$$

Following Lemma 5.4 in (Srinivas et al., 2010), $\sum_{t=1}^{T} 4\beta_T \sigma_t^2(\mathbf{x}_t, \mathbf{z}_t) \leq C_1 \beta_T \gamma_T$ where $C_1 \triangleq 8/\log(1 + \sigma_n^{-2})$. Hence,

$$
\begin{aligned}
4\beta_T \left( \sum_{t=1}^{T} \sigma_t(\mathbf{x}_t, \mathbf{z}_t) \right)^2 &\leq C_1 T \beta_T \gamma_T \\
2\beta_T^{1/2} \sum_{t=1}^{T} \sigma_t(\mathbf{x}_t, \mathbf{z}_t) &\leq \sqrt{C_1 T \beta_T \gamma_T} .
\end{aligned}
$$

As a result,

$$
R_T \leq (\eta(1 - \lambda) + 1) \sqrt{C_1 T \beta_t \gamma_T} .
$$

## E  PROOF OF THEOREM 3.7

Recall that $\mathbf{x}_t^- \triangleq \arg\max_{\mathbf{x} \in \mathcal{X}} \rho_{l_t}(\mathbf{x}; \alpha)$ and at iteration $t$, we recommend $\mathbf{x}_t^\star = \mathbf{x}_{\zeta(t)}^-$ where

$$
\zeta(t) \triangleq \arg\max_{t' \in \{1,\dots,t\}} \rho_{l_{t'}}(\mathbf{x}_{t'}^-; \alpha) . \tag{22}
$$

We need to prove that with probability $\geq 1 - \delta$,

$$
r(\mathbf{x}_t^\star) = r(\mathbf{x}_{\zeta(t)}^-) \leq \eta \sqrt{C_1 \beta_t \gamma_t / t} .
$$

*Proof.* From the definition of $\zeta(t)$ in equation (22),

$$
\rho_{l_{\zeta(t)}}(\mathbf{x}_{\zeta(t)}^-; \alpha) \geq \rho_{l_{t'}}(\mathbf{x}_{t'}^-; \alpha) \; \forall 1 \leq t' \leq t .
$$

Hence,

$$\rho_{l_{\zeta(t)}}(\mathbf{x}^-_{\zeta(t)}; \alpha) \geq \frac{1}{t} \sum_{t'=1}^{t} \rho_{l_{t'}}(\mathbf{x}^-_{t'}; \alpha) . \tag{23}$$

$$
\begin{aligned}
r(\mathbf{x}^-_{\zeta(t)}) &\triangleq \rho_f(\mathbf{x}_*; \alpha) - \rho_f(\mathbf{x}^-_{\zeta(t)}; \alpha) \\
&\leq \rho_f(\mathbf{x}_*; \alpha) - \rho_{l_{\zeta(t)}}(\mathbf{x}^-_{\zeta(t)}; \alpha) \\
&\leq \rho_f(\mathbf{x}_*; \alpha) - \frac{1}{t} \sum_{t'=1}^{t'} \rho_{l_{t'}}(\mathbf{x}^-_{t'}; \alpha) \\
&= \frac{1}{t} \sum_{t'=1}^{t} \left( \rho_f(\mathbf{x}_*; \alpha) - \rho_{l_{t'}}(\mathbf{x}^-_{t'}; \alpha) \right) \\
&\leq \frac{1}{t} \sum_{t'=1}^{t} \left( \rho_{u_{t'}}(\mathbf{x}_*; \alpha) - \rho_{l_{t'}}(\mathbf{x}^-_{t'}; \alpha) \right) \\
&\leq \frac{1}{t} \sum_{t'=1}^{t} \left( \rho_{u_{t'}}(\mathbf{x}^+_{t'}; \alpha) - \rho_{l_{t'}}(\mathbf{x}^-_{t'}; \alpha) \right) \\
&\leq \frac{1}{t} \sum_{t'=1}^{t} \min\left( \eta \left( \rho_{u_{t'}}(\mathbf{x}_{t'}; \alpha) - \rho_{l_{t'}}(\mathbf{x}_{t'}; \alpha) \right), \frac{1}{\lambda} \left( \rho_{u_{t'}}(\mathbf{x}_{t'}; \alpha) - \rho_{l_{t'}}(\mathbf{x}^-_{t'}; \alpha) \right) \right) \\
&\leq \frac{1}{t} \sum_{t'=1}^{t} \min\left( \eta \left( \rho_{u_{t'}}(\mathbf{x}_{t'}; \alpha) - \rho_{l_{t'}}(\mathbf{x}_{t'}; \alpha) \right), \frac{1}{\lambda} \left( \rho_{u_{t'}}(\mathbf{x}_{t'}; \alpha) - \rho_{l_{t'}}(\mathbf{x}_{t'}; \alpha) \right) \right) \\
&= \frac{1}{t} \sum_{t'=1}^{t} \eta \left( \rho_{u_{t'}}(\mathbf{x}_{t'}; \alpha) - \rho_{l_{t'}}(\mathbf{x}_{t'}; \alpha) \right) \\
&\leq \frac{1}{t} \sum_{t'=1}^{t} \eta \left( 2\beta_{t'}^{1/2} \sigma_{t'}(\mathbf{x}_{t'}, \mathbf{z}_{t'}) \right) \\
&\leq \frac{1}{t} \sum_{t'=1}^{t} \eta \left( 2\beta_{t}^{1/2} \sigma_{t'}(\mathbf{x}_{t'}, \mathbf{z}_{t'}) \right) \qquad \text{(from the monotonicity of } \beta_{t'}) \\
&\leq \frac{\eta}{t} \sqrt{C_1 t \beta_t \gamma_t} \\
&= \eta \sqrt{C_1 \beta_t \gamma_t / t} .
\end{aligned}
$$

(24), (25), (26), (27), (28), (29)

where equation (24) is from equation (23); equation (25) is from OC equation (6) and UC equation (7); equation (26) is because $\rho_{l_{t'}}(\mathbf{x}^-_{t'}; \alpha) \geq \rho_{l_{t'}}(\mathbf{x}_{t'}; \alpha)$; equation (27) is because $\eta \in [1, 1/\lambda]$; equation (28) is from equation (15); equation (29) is from equation (21) .

$\square$

## F  META-VBO FOR VANILLA BO

### F.1  BO WITH V-SET

In this section, we consider the vanilla Bayesian optimization (BO) of a blackbox function. The setting is similar to Sec. 2.1 except that there are not any environmental random variables.

In particular, let the blackbox function be denoted as $f : \mathcal{X} \to \mathbb{R}$. The evaluation of $f$ is expensive and we can only observe noisy evaluation $y(\mathbf{x}) \triangleq f(\mathbf{x}) + \epsilon(\mathbf{x})$ where $\epsilon(\mathbf{x}) \sim \mathcal{N}(0, \sigma_n^2)$. We would like to find the maximizer

$$\mathbf{x}_* \triangleq \arg\max_{\mathbf{x} \in \mathcal{X}} f(\mathbf{x}) . \tag{30}$$

BO is about designing a sequential strategy of selecting input query $\mathbf{x}_t$ at iteration $t$, considering observations in the previous $t-1$ iterations, denoted as $y_{\mathbf{D}_t} \triangleq (y(\mathbf{x}_{t'}))_{t'=1}^{t-1}$ where $\mathbf{D}_t \triangleq (\mathbf{x}_{t'})_{t'=1}^{t-1}$ to quickly discovery $\mathbf{x}_*$.

To obtain the posterior belief of $f$ given observations $\mathbf{y}_{\mathbf{D}_t}$, the blackbox function $f$ is often modelled with a GP (Rasmussen and Williams, 2006). Let the posterior belief of $f(\mathbf{x})$ be denoted as $\mathcal{N}(\mu_t(\mathbf{x}), \sigma_t^2(\mathbf{x}))$.

Under the assumption that $f$ belonging to the RKHS space with a norm bounded by $B$, we adopt the well-known confidence bound of $f(\mathbf{x})$ from the work of Chowdhury and Gopalan (2017). Let $\gamma_t$ denote the maximum information gain about $f$ that can be obtained from any set of $t-1$ observations. Choosing $\delta \in (0,1)$, $\beta_t = (B + \sigma_n\sqrt{2(\gamma_t + 1 + \log 1/\delta)})^2$, then the event that

$$\forall t \geq 1, \forall \mathbf{x} \in \mathcal{X},\ l_t(\mathbf{x}) \leq f(\mathbf{x}) \leq u_t(\mathbf{x}) \tag{31}$$

holds with probability $\geq 1 - \delta$ where the lower and upper confidence bounds are defined as

$$l_t(\mathbf{x}) \triangleq \mu_t(\mathbf{x}) - \beta_t^{1/2}\sigma_t(\mathbf{x}) \quad \text{and} \quad u_t(\mathbf{x}) \triangleq \mu_t(\mathbf{x}) + \beta_t^{1/2}\sigma_t(\mathbf{x}) \text{ , resp.} \tag{32}$$

It follows that

$$\forall t \geq 1,\ f(\mathbf{x}_*) \in [l_t(\mathbf{x}_t^-), u_t(\mathbf{x}_t^+)] \tag{33}$$

where $\mathbf{x}_t^- \triangleq \arg\max_{\mathbf{x} \in \mathcal{X}} l_t(\mathbf{x})$ and $\mathbf{x}_t^+ \triangleq \arg\max_{\mathbf{x} \in \mathcal{X}} u_t(\mathbf{x})$.

The *versatile query set* (V-set), denoted as $\mathcal{V}_t$, consists of those inputs $\mathbf{x} \in \mathcal{X}$ that satisfies the following overlapping condition (OC) and uncertainty condition (UC)

$$u_t(\mathbf{x}) \geq \max\left( \underbrace{l_t(\mathbf{x}_t^-) + \lambda(u_t(\mathbf{x}_t^+) - l_t(\mathbf{x}_t^-))}_{\text{overlapping condition}}, \underbrace{l_t(\mathbf{x}) + \frac{1}{\eta}\left(u_t(\mathbf{x}_t^+) - l_t(\mathbf{x}_t^-)\right)}_{\text{uncertainty condition}} \right) \tag{34}$$

where $\lambda \in [0,1]$ and $\eta \in [1, 1/\lambda]$. Similar to the proof of Theorem 3.2, we can obtain the following theorem on the cumulative regret of selecting the input query in V-set $\mathcal{V}_t$.

**Theorem F.1.** *Given the selection of $\mathbf{x}_t$ as any input in $\mathcal{V}_t$, the algorithm achieves a sublinear cumulative regret:*

$$\forall T \geq 1, R_T \leq (\eta(1-\lambda) + 1)\sqrt{C_1 T \beta_T \gamma_T} \tag{35}$$

*holds with probability $\geq 1 - \delta$ where $C_1 \triangleq 8/\log(1 + \sigma_n^{-2})$; $\gamma_T$, $\beta_T$, $\delta$ are defined in equation (31); $\lambda$ and $\eta$ are defined in equation (34).*

### F.2 META-VBO

Adopting the meta-BO setting in Sec. 3, we consider additional prior tasks $\tau \in \mathcal{T}$. The blackbox function of any prior task $\tau$, denoted as $f_\tau : \mathcal{X} \to \mathbb{R}$, is assumed to belong to an RKHS with a norm bounded by $B$. The optimization result of a prior task $\tau$ includes the GP posterior mean and variance of $f_\tau$, denoted as $\hat{\mu}_\tau : \mathcal{X} \to \mathbb{R}$ and $\hat{\sigma}_\tau^2 : \mathcal{X} \to \mathbb{R}^+$, respectively. We denote the lower and upper bound of the function evaluation $f_\tau(\mathbf{x})$ as $[\hat{l}_\tau(\mathbf{x}), \hat{u}_\tau(\mathbf{x})]$ where

$$\hat{l}_\tau(\mathbf{x}) \triangleq \hat{\mu}_\tau(\mathbf{x}) - \beta_{t_\tau}^{1/2}\hat{\sigma}_\tau(\mathbf{x}) , \qquad \hat{u}_\tau(\mathbf{x}) \triangleq \hat{\mu}_\tau(\mathbf{x}) + \beta_{t_\tau}^{1/2}\hat{\sigma}_\tau(\mathbf{x}) \tag{36}$$

and $t_\tau$ is the iteration where the BO procedure on prior task $\tau$ stops.

From Sec. 3.2, we utilize the prior tasks to assign different *priorities* to inputs in $\mathcal{V}_t$. We can approach this in two distinct ways.

1. We assume that the global maximizer of the current task is close to a *local maximizer* of a prior task. In particular, we assign higher priorities to inputs that have a higher chance of being a local maximizer (restricted to $\mathcal{V}_t$) of prior tasks. Under this assumption, we introduce the notion of *the probable local maximizer set*, denoted as $\mathcal{P}_{\tau|\mathcal{V}_t}$, in a similar manner to that in Sec. 3.2:

$$\mathcal{P}_{\tau|\mathcal{V}_t} \triangleq \{\mathbf{x} \in \mathcal{V}_t |\ \hat{u}_\tau(\mathbf{x}) \geq \max_{\mathbf{x}' \in \mathcal{V}_t} \hat{l}_\tau(\mathbf{x}')\} . \tag{37}$$

2. We assume that the global maximizer of the current task is close to a *global maximizer* of a prior task. In particular, we assign higher priorities to inputs that have a higher chance of being a global maximizer of prior tasks. Under this assumption, we introduce the notion of *the probable global maximizer set*, denoted as $\mathcal{P}_{\tau|\mathcal{X}}$, in a similar manner to that in Sec. 3.2:

$$\mathcal{P}_{\tau|\mathcal{X}} \triangleq \{\mathbf{x} \in \mathcal{X} | \, \widehat{u}_\tau(\mathbf{x}) \geq \max_{\mathbf{x}' \in \mathcal{X}} \widehat{l}_\tau(\mathbf{x}')\} \, . \tag{38}$$

Let us denote the probable maximizer set as $\mathcal{P}_\tau$ which is either the probable local maximizer set $\mathcal{P}_{\tau|\mathcal{V}_t}$ or the probable global maximizer set $\mathcal{P}_{\tau|\mathcal{X}}$ depending on the assumption one may adopt. Then, the priority of an input $\mathbf{x} \in \mathcal{V}_t$, denoted as $\pi_t(\mathbf{x})$, is measured by the number of the probable maximizer sets that $\mathbf{x}$ belongs to, i.e.,

$$\forall \mathbf{x} \in \mathcal{V}_t, \ \pi_t(\mathbf{x}) \triangleq \sum_{\tau \in \mathcal{T}} \mathbb{1}_{\mathbf{x} \in \mathcal{P}_\tau} \, . \tag{39}$$

The higher $\pi_t(\mathbf{x})$ is, the more likely that $\mathbf{x}$ is in the probable maximizer set of a prior task. Hence, we select the input query $\mathbf{x}_t$ as

$$\mathbf{x}_t \triangleq \max_{\mathbf{x} \in \mathcal{V}_{t,\max}} u_t(\mathbf{x}) \ \text{where} \ \mathcal{V}_{t,\max} \triangleq \{\mathbf{x} \in \mathcal{V}_t | \, \pi_t(\mathbf{x}) = \max_{\mathbf{x}' \in \mathcal{V}_t} \pi_t(\mathbf{x}')\} \, . \tag{40}$$

Following Sec. 3.3, we propose to recommend $\mathbf{x}_t^* \triangleq \mathbf{x}_{\zeta(t)}^-$ as an approximation to the optimal solution where $\zeta(t) \triangleq \arg\max_{t' \in \{1,\ldots,t\}} l_{t'}(\mathbf{x}_{t'}^-)$.

**Theorem F.2.** *Let* $\zeta(t) \triangleq \arg\max_{t' \in \{1,\ldots,t\}} l_{t'}(\mathbf{x}_{t'}^-)$, *then, by recommending* $\mathbf{x}_t^\star = \mathbf{x}_{\zeta(t)}^-$, *the instantaneous regret at* $\mathbf{x}_t^\star$ *is bounded:* $r(\mathbf{x}_t^\star) \leq \eta\sqrt{C_1\beta_t\gamma_t/t}$ *with probability* $\geq 1 - \delta$.

# G  META-VBO FOR ADVERSARIALLY ROBUST BAYESIAN OPTIMIZATION

Let us consider the adversarially robust BO problem from the work of Bogunovic et al. (2018) that is reformulated in the work of Nguyen et al. (2021b). It is to find the maximizer $\mathbf{x}_* \triangleq \arg\max_{\mathbf{x} \in \mathcal{X}} \min_{\mathbf{z} \in \mathcal{Z}} f(\mathbf{x}, \mathbf{z})$. Furthermore, Nguyen et al. (2021b) casts this problem as a BO of VaR problem for $\alpha \to 0^+$ and a uniform distribution of $\mathbf{Z}$. Let us denote $v_f(\mathbf{x}; 0^+) \triangleq \min_{\mathbf{z} \in \mathcal{Z}} f(\mathbf{x}, \mathbf{z})$. Then, by replacing $\rho_f(\mathbf{x}; \alpha)$ with $v_f(\mathbf{x}; 0^+)$, equivalently, $\min_{\mathbf{z} \in \mathcal{Z}} f(\mathbf{x}, \mathbf{z})$, we obtain a meta-BO solution to the adversarially robust BO.

Specifically, from the following confidence bounds of $\min_{\mathbf{z} \in \mathcal{Z}} f(\mathbf{x}, \mathbf{z})$

$$\min_{\mathbf{z} \in \mathcal{Z}} l_t(\mathbf{x}, \mathbf{z}) \leq \min_{\mathbf{z} \in \mathcal{Z}} f(\mathbf{x}, \mathbf{z}) \leq \min_{\mathbf{z} \in \mathcal{Z}} u_t(\mathbf{x}, \mathbf{z}) \, ,$$

we obtain the confidence bounds of $\min_{\mathbf{z} \in \mathcal{Z}} f(\mathbf{x}_*, \mathbf{z})$:

$$\min_{\mathbf{z} \in \mathcal{Z}} l_t(\mathbf{x}_t^-, \mathbf{z}) \leq \min_{\mathbf{z} \in \mathcal{Z}} f(\mathbf{x}_*, \mathbf{z}) \leq \min_{\mathbf{z} \in \mathcal{Z}} u_t(\mathbf{x}_t^+, \mathbf{z})$$

where $\mathbf{x}_t^- \triangleq \arg\max_{\mathbf{x} \in \mathcal{X}} \min_{\mathbf{z} \in \mathcal{Z}} l_t(\mathbf{x}, \mathbf{z})$ and $\mathbf{x}_t^+ \triangleq \arg\max_{\mathbf{x} \in \mathcal{X}} u_t(\mathbf{x}, \mathbf{z})$.

The V-set $\mathcal{V}_t$ is defined as the set of inputs $\mathbf{x} \in \mathcal{X}$ that satisfy:

- Overlapping condition (OC):

$$\min_{\mathbf{z} \in \mathcal{Z}} u_t(\mathbf{x}, \mathbf{z}) \geq \min_{\mathbf{z} \in \mathcal{Z}} l_t(\mathbf{x}_t^-, \mathbf{z}) + \lambda(\min_{\mathbf{z} \in \mathcal{Z}} u_t(\mathbf{x}_t^+, \mathbf{z}) - \min_{\mathbf{z} \in \mathcal{Z}} l_t(\mathbf{x}_t^-, \mathbf{z}))$$

where $\lambda \in [0, 1]$.

- Uncertainty condition (UC):

$$\min_{\mathbf{z} \in \mathcal{Z}} u_t(\mathbf{x}, \mathbf{z}) - \min_{\mathbf{z} \in \mathcal{Z}} l_t(\mathbf{x}, \mathbf{z}) \geq \frac{1}{\eta}(\min_{\mathbf{z} \in \mathcal{Z}} u_t(\mathbf{x}_t^+, \mathbf{z}) - \min_{\mathbf{z} \in \mathcal{Z}} l_t(\mathbf{x}_t^-, \mathbf{z}))$$

where $\eta \in [1, 1/\lambda]$.

---

**Algorithm 1** Meta-BO of Risk Measure $\rho$ and risk level $\alpha$

---

**Require:** $\mathcal{X}, \alpha, \lambda, \eta, (\mathbf{D}_1, \mathbf{y}_{\mathbf{D}_1}), (\rho_{\widehat{l}_\tau}, \rho_{\widehat{u}_\tau})_{\tau \in \mathcal{T}}$

1: Compute GP posterior given $(\mathbf{D}_1, \mathbf{y}_{\mathbf{D}_1})$: $\mu_1, \sigma_1$.
2: **for** $t$ in $1, 2, \ldots, T$ **do**
3:      Compute $\phi_t^- \triangleq \rho_{l_t}(\mathbf{x}_t^-; \alpha)$ and $\phi_t^+ \triangleq \rho_{u_t}(\mathbf{x}_t^+; \alpha)$.
4:      Compute the uncertainty of $\rho_f(\mathbf{x}_*; \alpha)$: $\mathrm{CI}_* \triangleq \phi_t^+ - \phi_t^-$.
5:      Find $\mathcal{V}_t \triangleq \{\mathbf{x} \in \mathcal{X} | \rho_{u_t}(\mathbf{x}; \alpha) \geq \phi_t^- + \lambda \mathrm{CI}_* \wedge \rho_{u_t}(\mathbf{x}; \alpha) - \rho_{l_t}(\mathbf{x}; \alpha) \geq \eta^{-1} \mathrm{CI}_*\}$.
6:      For $\tau \in \mathcal{T}$, compute $\phi_{\tau,t}^- \triangleq \max_{\mathbf{x} \in \mathcal{V}_t} \rho_{\widehat{l}_\tau}(\mathbf{x}; \alpha)$.
7:      Select $\mathbf{x}_t = \arg\max_{\mathbf{x} \in \mathcal{V}_t} \left( \sum_{\tau \in \mathcal{T}} \mathbb{1}_{\rho_{\widehat{u}_\tau}(\mathbf{x}; \alpha) \geq \phi_{\tau,t}^-}, \rho_{u_t}(\mathbf{x}; \alpha) \right)$.
8:      Select $\mathbf{z}_t$ as a lacing value w.r.t. $\mathbf{x}_t, l_t, u_t$, and $\alpha$ (Nguyen et al., 2021b;a).
9:      $\mathbf{D}_{t+1} = \mathbf{D}_t \cup \{(\mathbf{x}_t, \mathbf{z}_t)\}$ and $\mathbf{y}_{\mathbf{D}_{t+1}} = \mathbf{y}_{\mathbf{D}_t} \cup \{y_t(\mathbf{x}_t, \mathbf{z}_t)\}$.
10:      Compute GP posterior given $(\mathbf{D}_{t+1}, \mathbf{y}_{\mathbf{D}_{t+1}})$: $\mu_{t+1}$ and $\sigma_{t+1}$.
11: **end for**

---

Given $\mathbf{x}_t$ chosen as any input from $\mathcal{V}_t$, we select $\mathbf{z}_t = \arg\min_{\mathbf{z} \in \mathcal{Z}} l_t(\mathbf{x}_t, \mathbf{z})$ (Bogunovic et al., 2018). This choice of $\mathbf{z}_t$ is a lacing value (LV) w.r.t. $\mathbf{x}_t, l_t, u_t$, and $\alpha \to 0^+$ since it satisfies the definition of LV (Nguyen et al., 2021b):

$$l_t(\mathbf{x}_t, \mathbf{z}_t) = \min_{\mathbf{z} \in \mathcal{Z}} l_t(\mathbf{x}_t, \mathbf{z}) \leq \min_{\mathbf{z} \in \mathcal{Z}} u_t(\mathbf{x}_t, \mathbf{z}) \leq u_t(\mathbf{x}_t, \mathbf{z}_t)$$

The priorities assigned to inputs $\mathcal{V}_t$ follow our discussion in Sec. 3.2. In particular, the probable local maximizer set is constructed as follows.

$$\mathcal{P}_{\tau|\mathcal{V}_t} = \{\mathbf{x} \in \mathcal{V}_t | \min_{\mathbf{z} \in \mathcal{Z}} \widehat{u}_\tau(\mathbf{x}, \mathbf{z}) \geq \max_{\mathbf{x}' \in \mathcal{V}_t} \min_{\mathbf{z} \in \mathcal{Z}} \widehat{l}_\tau(\mathbf{x}', \mathbf{z})\}.$$

Then, the input query $\mathbf{x}_t$ is selected as

$$\mathbf{x}_t = \max_{\mathbf{x} \in \mathcal{V}_{t,\max}} \min_{\mathbf{z} \in \mathcal{Z}} u_t(\mathbf{x}, \mathbf{z})$$

where $\mathcal{V}_{t,\max}$ is defined in Sec. 3.2.

Regarding the recommended input, we follow the recommendation strategy in Sec. 3.3 to choose $\mathbf{x}_t^\star = \mathbf{x}_{\zeta(t)}^-$ where $\zeta(t) = \arg\max_{t' \in \{1, \ldots, t\}} \min_{\mathbf{z} \in \mathcal{Z}} l_{t'}(\mathbf{x}_{t'}^-, \mathbf{z})$.

## H  PSEUDOCODE

The pseudocode of our proposed algorithm is shown in Algorithm 1. It is noted that $\rho_{\widehat{l}_\tau}$ and $\rho_{\widehat{u}_\tau}$ represent functions for calculating the lower and upper bounds of the risk measure, respectively. In line 7, the comparison between tuples $\left( \sum_{\tau \in \mathcal{T}} \mathbb{1}_{\rho_{\widehat{u}_\tau}(\mathbf{x}; \alpha) \geq \phi_{\tau,t}^-}, \rho_{u_t}(\mathbf{x}; \alpha) \right)$ is executed elementwise. This comparison involves assessing the second position if the first position results in a tie, akin to the comparison operations in Python.

We demonstrate that the additional computational complexity of our proposed algorithm is not significant when compared to V-UCB, which does not utilize any prior tasks. Assuming finite domains $\mathcal{X}$ and $\mathcal{Z}$ to leverage the computational complexity presented in Nguyen et al. (2021b), each iteration of V-UCB has a complexity of $\mathcal{O}(|\mathcal{Z}||\mathcal{X}|(|\mathbf{D}_t|^2 + \log|\mathcal{Z}|))$ (Nguyen et al., 2021b). Both lines 3 and 5 in our proposed Algorithm 1 share the same computational complexity of $\mathcal{O}(|\mathcal{Z}||\mathcal{X}|(|\mathbf{D}_t|^2 + \log|\mathcal{Z}|))$ as it requires the computation of VaR as in V-UCB. Line 6, involving iteration over prior tasks, incurs an additional $\mathcal{O}(|\mathcal{T}||\mathcal{Z}||\mathcal{X}|(|\mathbf{D}_t|^2 + \log|\mathcal{Z}|))$. Consequently, compared to V-UCB, our algorithm introduces only an extra linear dependence on the number of prior tasks.

## I  ADDITIONAL EXPERIMENTS

### I.1  ILLUSTRATION OF VERSATILE QUERY SET AND PROBABLE LOCAL MAXIMIZER SET

We visualize the versatile query set (V-set) $\mathcal{V}_t$ and the probable local maximizer sets $\mathcal{P}_{\tau|\mathcal{V}_t}$ when we maximizer VaR of a blackbox function given a set of prior tasks in 3 different scenarios.

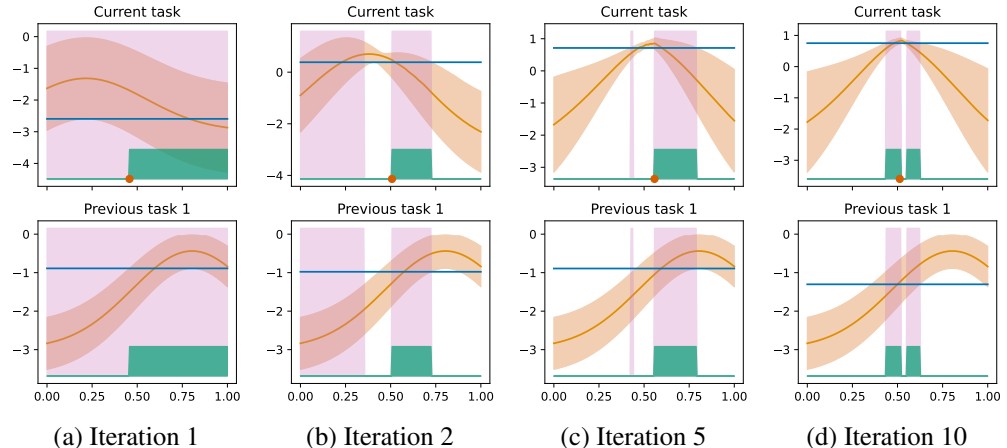

(a) Iteration 1       (b) Iteration 2       (c) Iteration 5       (d) Iteration 10

Figure 5: Visualization of the V-set $\mathcal{V}_t$ and the probable local maximizer sets $\mathcal{P}_{\tau|\mathcal{V}_t}$ when the prior task has poorly-estimated VaR. The figures in each column share the same $x$-axis. The top figure displays the GP posterior mean (the orange line) along with the lower and upper bounds (the orange shaded area) for the blackbox function of the current task. The bottom figure displays the same information for the prior tasks. In both figures, the blue line indicates the maximum lower bound, while the pink shaded area represents $\mathcal{V}_t$. Although $\mathcal{V}_t$ is defined based on the bounds of the blackbox function of the current task, it is also plotted in the bottom figures to aid the visualization of the probable local maximizer set $\mathcal{P}_{\tau|\mathcal{V}_t}$. While the green shaded area in the bottom figure shows the probable local maximizer set $\mathcal{P}_{\tau|\mathcal{V}_t}$ for the prior task, in the top figure, it shows the count of probable local maximizer sets to which an input belongs (a higher green area corresponds to a larger number of probable local maximizer sets). The red dot shows the input query $\mathbf{x}_t$.

1. **Figure 5: Prior task with poorly-estimated VaR.** There is a single task that differs from the current task. Furthermore, VaR of the prior task is not well-estimated due to sparse observations in the prior task, as shown by the considerable gap between the lower and upper bounds of VaR in the bottom figures. During the initial BO iterations, when VaR of the current task remains uncertain across the input domain, the prior task steers the input query towards the region where VaR of prior tasks is likely to attain high evaluations. However, as the posterior distribution of VaR in the current task refines (e.g., see iteration 10), the influence of the prior task on the input query diminishes. Hence, we observe that as $\mathcal{V}_t$ reduces, the estimation of VaR in the prior task becomes inaccurate to pinpoint promising inputs within $\mathcal{V}_t$. It is noted that at iteration 2 (Figure 5b), the gap between pink shaded regions is due to the uncertainty condition in Equation equation (7).

2. **Figure 6: Prior task with well-estimated VaR.** In this scenario, the blackbox function in the prior task is similar to that in Figure 5. However, the estimation of VaR for the prior task is more accurate, indicated by a smaller gap between lower and upper bounds of VaR in the bottom figures. This is due to an increased number of observations in the prior task. With a well-estimated VaR in the prior task, the probable local maximizer sets $\mathcal{P}_{\tau|\mathcal{V}_t}$ (the green shaded areas) are small even at the first BO iteration. However, as the V-set $\mathcal{V}_t$ reduces, the influence of the prior task on the current optimization task reduces (e.g., see iteration 15).

3. **Figure 7: Multiple prior tasks.** There are 3 prior tasks: both prior tasks 1 & 3 differ from the current task while prior task 2 shares similarites in a local region around the maximizer; prior task 3 bears similarity to the current task. The varying heights of the green shaded areas in the top figures demonstrate the different priorities assigned to inputs within $\mathcal{V}_t$. In particular, this illustration shows that a substantial proportion of useful prior tasks leads to a priority assignment that identifies promising inputs within $\mathcal{V}_t$.

## I.2 EXPERIMENTS WITH 9-DIMENSIONAL INPUTS

In this experiment, we empirically demonstrate the effectiveness of our algorithm in optimizing VaR and CVaR for a blackbox function with 9-dimensional inputs. The blackbox function is a sum of

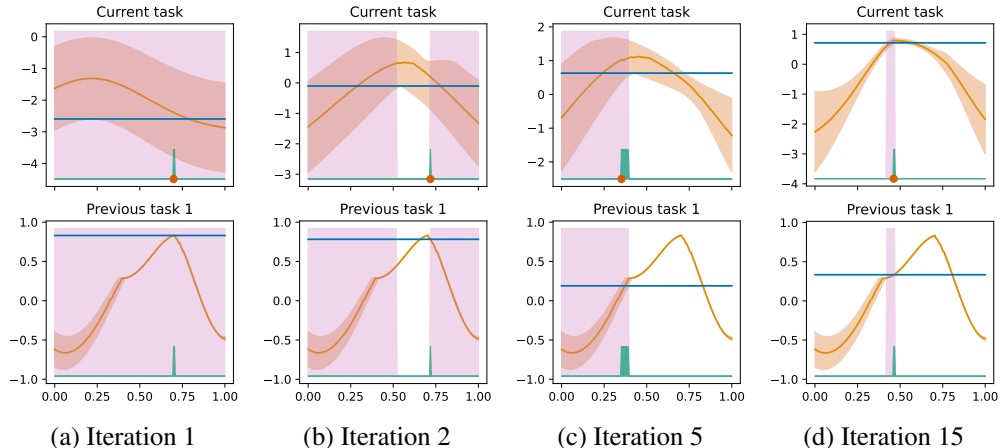

Figure 6: Visualization of the query set $\mathcal{V}_t$ and the probable local maximizer sets $\mathcal{P}_{\tau|\mathcal{V}_t}$ when the prior task has well-estimated VaR. The plot notations are similar to those in Figure 5. In this context, there exists a single prior task where the VaR maximizer is positioned around $x = 0.75$, contrasting with the VaR maximizer of the current task, which is situated approximately at $x = 0.5$.

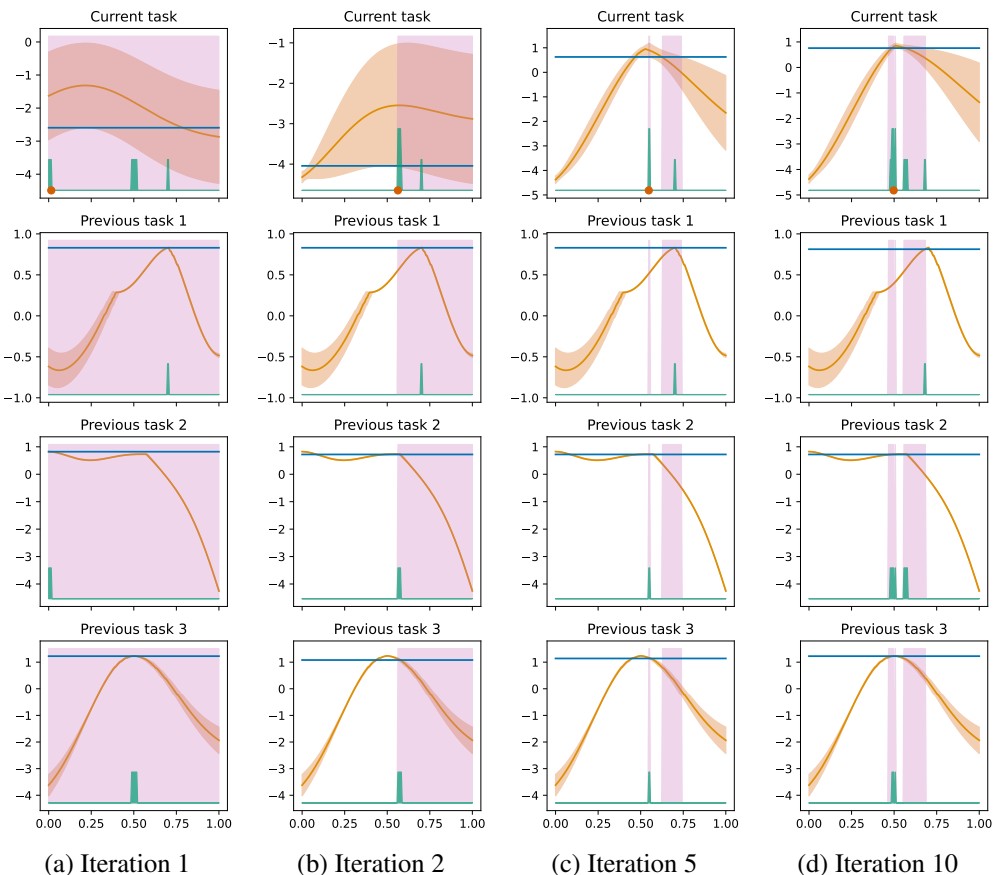

Figure 7: Visualization of the query set $\mathcal{V}_t$ and the probable local maximizer sets $\mathcal{P}_{\tau|\mathcal{V}_t}$ when there are multiple prior tasks. The plot notations are similar to those in Figure 5.

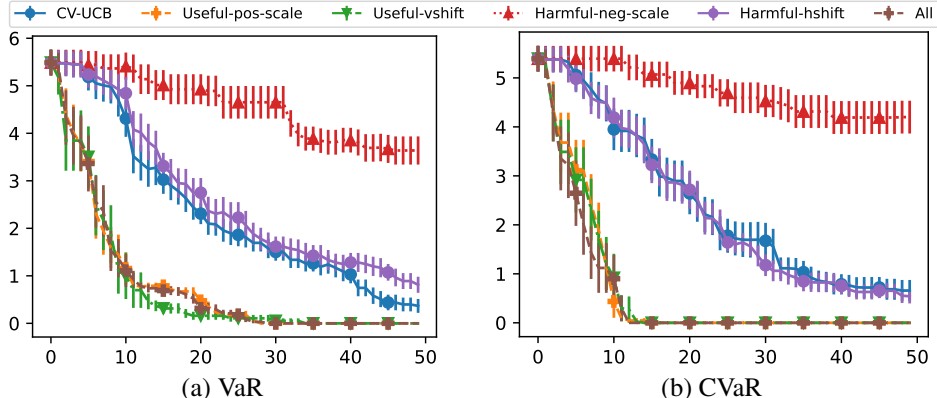

Figure 8: Plots of the regret at the recommended input against the BO iteration in optimizing (a) VaR and (b) CVaR of a blackbox function with 9-dimensional inputs.

three instances of the Hartmann-3D function. Our experiments involve a diverse set $\mathcal{T}$ of prior tasks, incorporating transformations such as positive and negative scaling, horizontal and vertical shifting, as detailed in Section 4. The outcomes are illustrated in Figure 8. When $\mathcal{T}$ comprises only useful tasks from $\mathcal{T}_{\texttt{useful-pos-scale}}$ and $\mathcal{T}_{\texttt{useful-hshift}}$, our proposed meta-VBO outperforms V-UCB (in Figure 8a) and CV-UCB (in Figure 8b). In the extreme scenario when $\mathcal{T}$ contains harmful tasks from $\mathcal{T}_{\texttt{harmful-neg-scale}}$ and $\mathcal{T}_{\texttt{harmful-hshift}}$, our algorithm may not surpass V-UCB or CV-UCB.

## I.3 Conditional Value-at-Risk Experiments

The Gaussian curve $f_{\text{Gaussian}}(x, z)$ is defined as follows.

$$f_{\text{Gaussian}}(x, z) = 5 \exp\left(-\begin{pmatrix} x \\ z \end{pmatrix}^\top \begin{pmatrix} 0.05 & 0 \\ 0 & 0.5 \end{pmatrix} \begin{pmatrix} x \\ z \end{pmatrix}\right) \tag{41}$$

where $\mathcal{X} \times \mathcal{Z}$ consists of 10000 data points in $[0, 1]^2$. The distribution of $Z$ is $\mathcal{N}(0.5, 0.09)$.

Other synthetic functions are obtained from https://www.sfu.ca/~ssurjano. The domains $\mathcal{X} \times \mathcal{Z}$ of the Branin-Hoo, the Goldstein-Price, the six-hump camel consist of 10000 data points in $[0, 1]^2$. The distributions of $\mathbf{Z}$ in these experiments are $\mathcal{N}(0.1, 0.2)$, $\mathcal{N}(0.1, 0.4)$, and $\mathcal{N}(0.1, 0.2)$, respectively. The domain $\mathcal{X} \times \mathcal{Z}$ of the Hartmann-3D function consists of 20000 data points in $[0, 1]^3$, and that of the Hartmann-6D function consists of 40000 data points in $[0, 1]^6$. The distributions of $\mathbf{Z}$ in the Hartmann-3D and Hartmann-6D experiments are $\mathcal{N}(0.9, 0.2)$ and $\mathcal{N}(0.5, 0.4)$, respectively. The portfolio optimization and the yacht hydrodynamics experiments are adopted from the works of Cakmak et al. (2020); Nguyen et al. (2021a). The domains $\mathcal{X} \times \mathcal{Z}$ of the two experiments consist of 30000 and 40000 data points, respectively. While the distribution of $\mathbf{Z}$ is uniform in the portfolio optimization experiment, it is $\mathcal{N}(0.5, 0.1)$ in the yacht hydrodynamics experiment. Figs. 9-14 show the current task and several prior tasks in the experiments with the Branin-Hoo, the Goldstein-Price, and the six-hump camel functions. While prior tasks in $\mathcal{T}_{\texttt{useful-vshift}}$ (i.e., vertical shifting) and $\mathcal{T}_{\texttt{useful-pos-scale}}$ (i.e., positive scaling) share common maximizers of VaR and CVaR of the current task, those in $\mathcal{T}_{\texttt{harmful-hshift}}$ (i.e., horizontal-**x**-shifting) and $\mathcal{T}_{\texttt{harmful-neg-scale}}$ (i.e., negative scaling) differ that of the current task.

Fig. 15 shows the average the standard error of $r(\mathbf{x}_t^\star)$ over 30 random repetitions of the BO of CVaR experiments. These experiments share the same sets of prior tasks with the BO of VaR experiments in Sec. 4: $\mathcal{T}_{\texttt{useful-pos-scale}}$, $\mathcal{T}_{\texttt{useful-vshift}}$, $\mathcal{T}_{\texttt{harmful-neg-scale}}$, $\mathcal{T}_{\texttt{harmful-hshift}}$, and $\mathcal{T}_{\texttt{all}}$. The risk level is set at $\alpha = 0.1$ for all experiments. We choose $\lambda = 0$ and $\eta = 1$ as explained in Sec. 3.3. In Fig. 15, we observe that when $\mathcal{T}$ contains only prior useful tasks in $\mathcal{T}_{\texttt{useful-pos-scale}}$ and $\mathcal{T}_{\texttt{useful-vshift}}$, our proposed meta-VBO outperforms CV-UCB (Nguyen et al., 2021a) which does not utilize any prior tasks. Furthermore, when $\mathcal{T}$ contains both harmful and useful tasks in $\mathcal{T}_{\texttt{all}}$, our meta-BO of CVaR is still able to exploit prior useful tasks and is robust against prior harmful tasks to outperforms CV-UCB. When $\mathcal{T}$ consists of only harmful tasks in $\mathcal{T}_{\texttt{harmful-neg-scale}}$

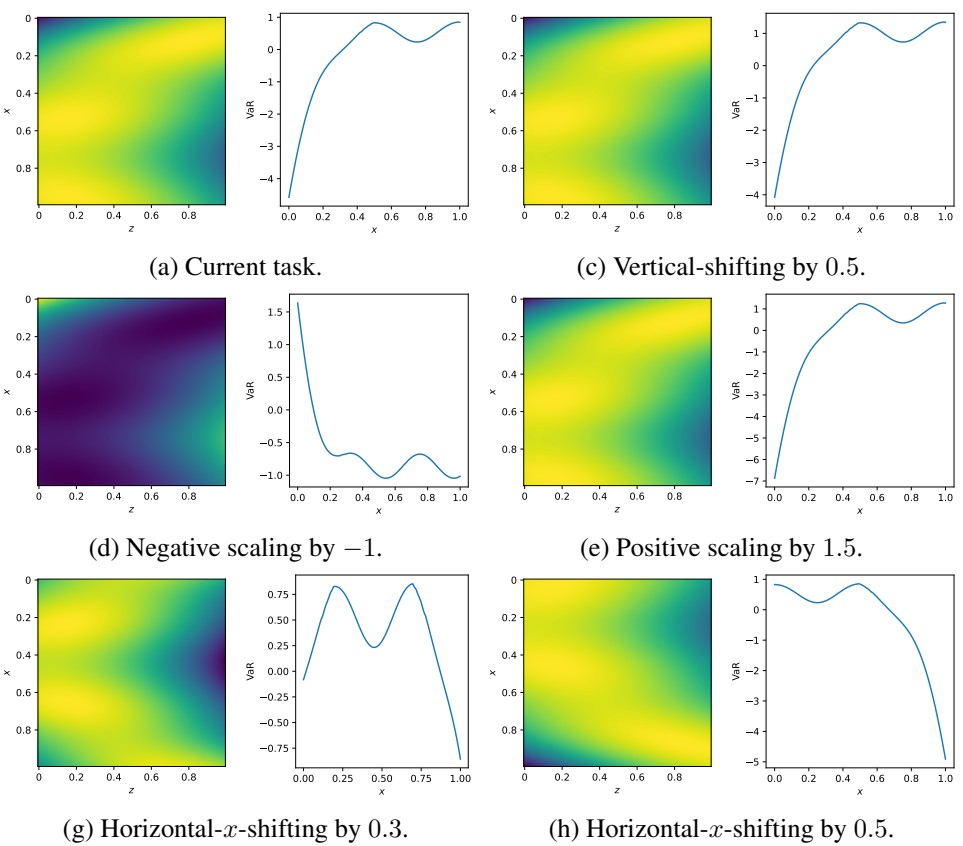

Figure 9: Current and prior BO of VaR tasks in Branin-Hoo experiments. The heatmaps show the blackbox function $f(\mathbf{x}, \mathbf{z})$.

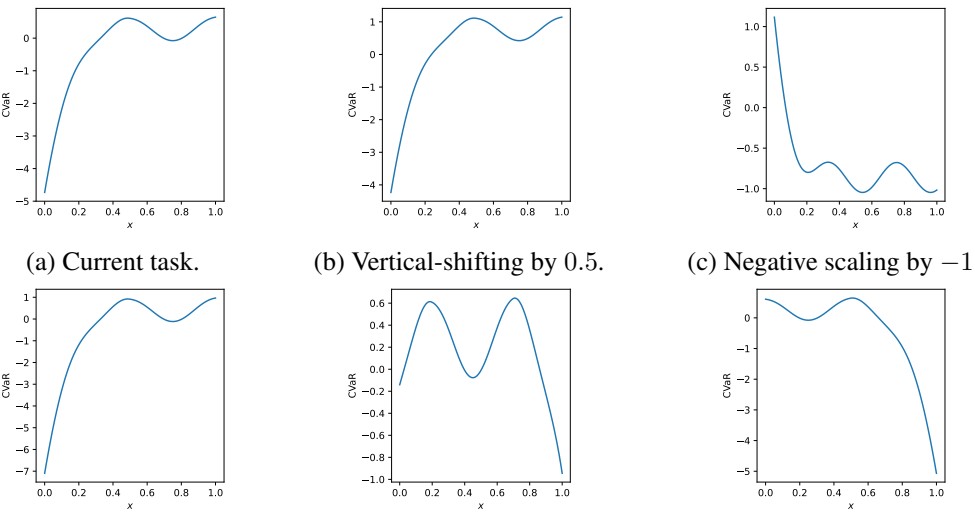

Figure 10: Current and prior BO of CVaR tasks in Branin-Hoo experiments.

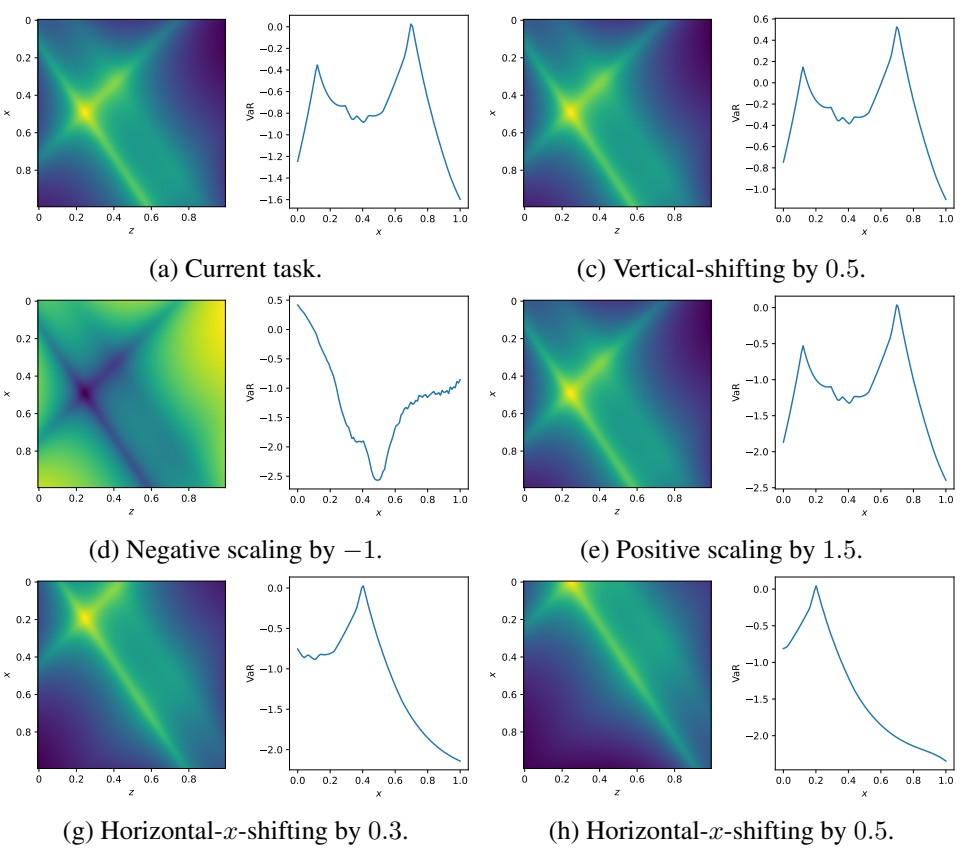

(a) Current task.

(c) Vertical-shifting by 0.5.

(d) Negative scaling by −1.

(e) Positive scaling by 1.5.

(g) Horizontal-$x$-shifting by 0.3.

(h) Horizontal-$x$-shifting by 0.5.

Figure 11: Current and prior BO of VaR tasks in Goldstein-Price experiments. The heatmaps show the blackbox function $f(\mathbf{x}, \mathbf{z})$.

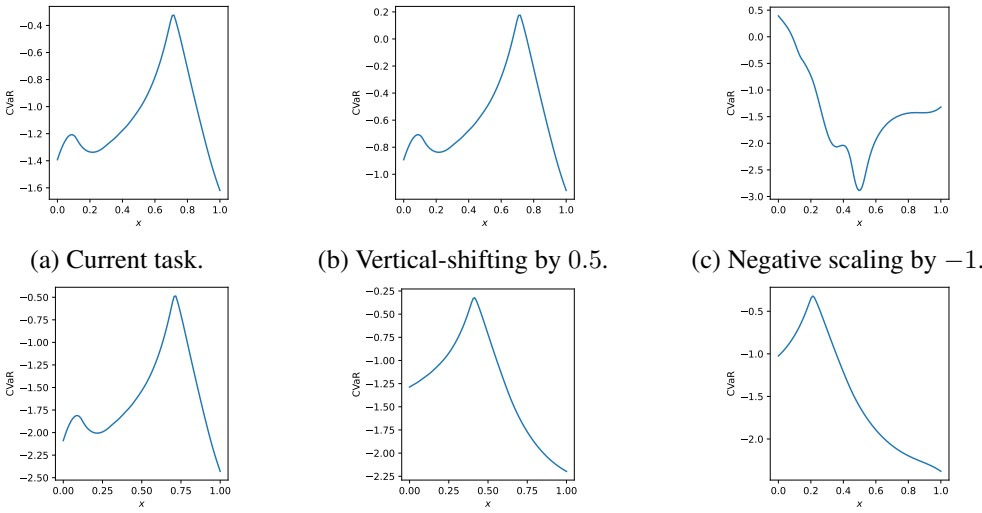

(a) Current task.

(b) Vertical-shifting by 0.5.

(c) Negative scaling by −1.

(d) Positive scaling by 1.5.

(e) Horizontal-$x$-shifting by 0.3.

(f) Horizontal-$x$-shifting by 0.5.

Figure 12: Current and prior BO of CVaR tasks in Goldstein-Price experiments.

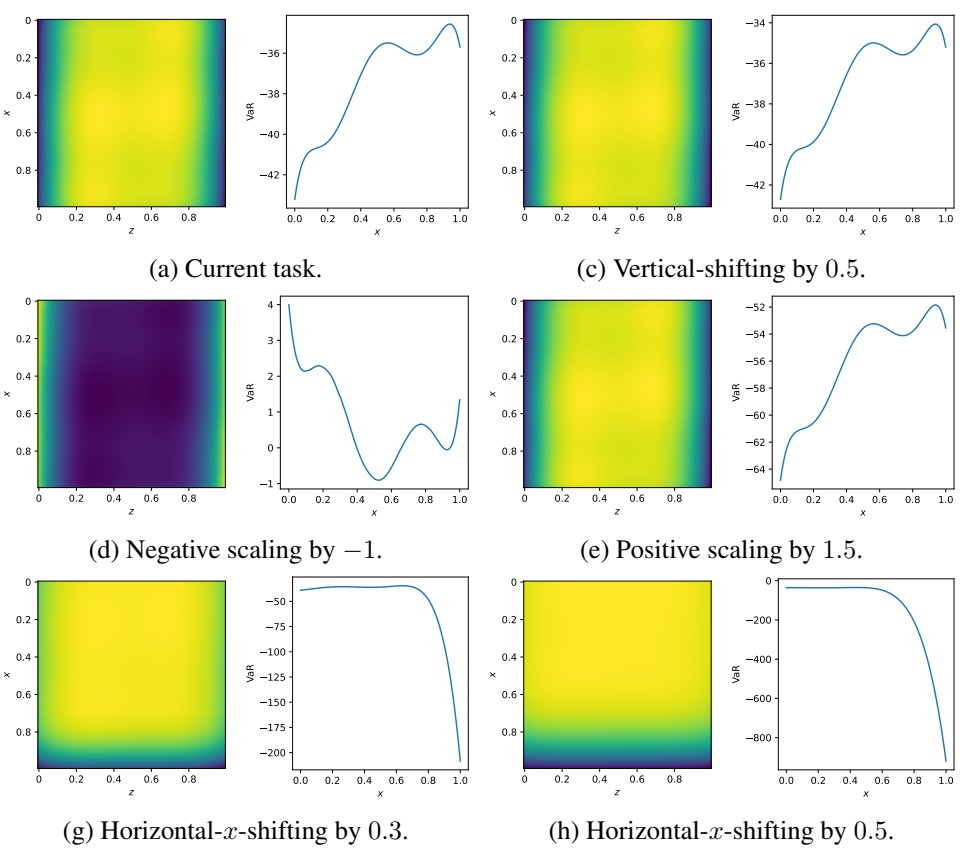

Figure 13: Current and prior BO of VaR tasks in six-hump camel experiments. The heatmaps show the blackbox function $f(\mathbf{x}, \mathbf{z})$.

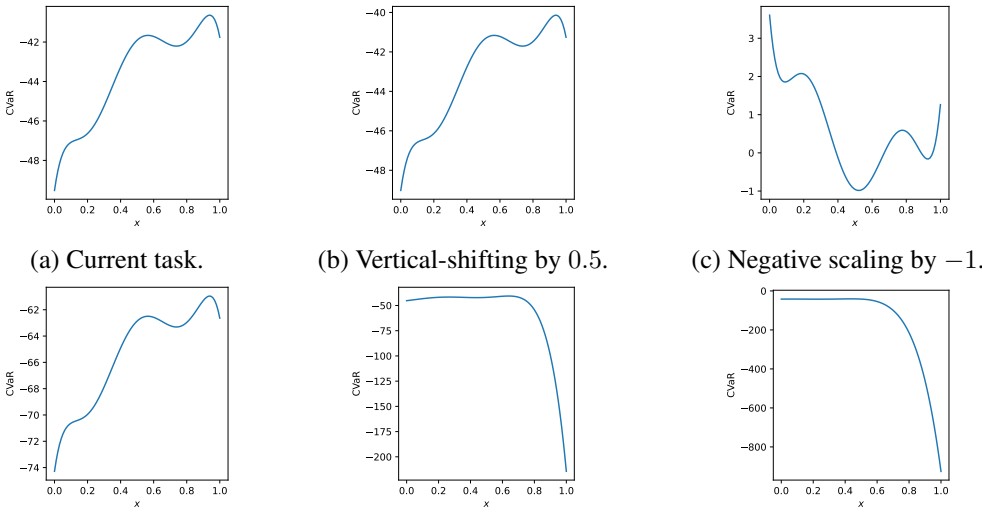

Figure 14: Current and prior BO of CVaR tasks in six-hump camel experiments.

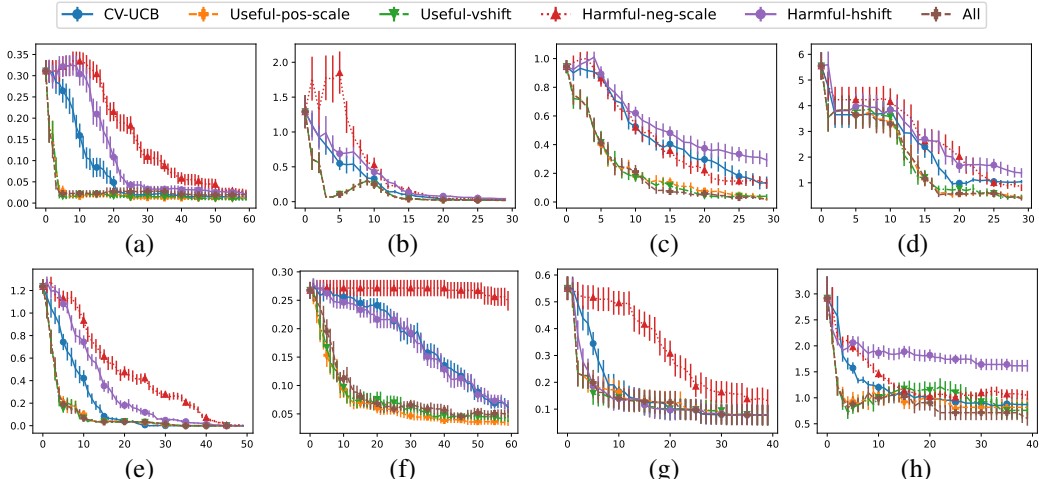

Figure 15: Meta-BO of CVaR experiments on (a) a Gaussian curve, (b) the Branin-Hoo function, (c) the Goldstein-Price function, (d) the six-hump camel function, (e) the Hartmann-3D function, (f) the Hartmann-6D function, (g) the yacht hydrodynamics dataset, and (h) the portfolio optimization dataset.

and $\mathcal{T}_{\texttt{hamrful-hsfhit}}$, meta-VBO does not outperform CV-UCB, but it still converges to a low-regret solution except for the experiment with the Hartmann-6D function. It is noted that the curves of `useful-pos-scale`, `useful-vshift`, and `all` overlap each other in most of the plots in Fig. 15.