# OpenReview forum: "Meta-VBO: Utilizing Prior Tasks in Optimizing Risk Measures with Gaussian Processes"
_ICLR.cc/2024/Conference — ICLR 2024 poster_

### Official Review · Reviewer_ZY9Q · 2023-10-26

**Soundness:** 4 excellent
**Presentation:** 4 excellent
**Contribution:** 3 good
**Rating:** 8
**Confidence:** 4

**Summary:**

The paper introduces a novel meta-learning based approach for Bayesian Optimization of risk measures. Given a set of previous tasks, the authors propose to construct a no-regret query set and they prove that any strategy picking inputs in such sets leads is a no-regret strategy, i.e., it has a sublinear regret. Furthermore, the authors assigns priorities on all the inputs belonging to this NQS $\mathcal{Q}_t$ at iteration $t$. Those importance weights are computed using the information from previous tasks. They take high values for the most probable local maximizers among the set of previous tasks. Then the authors introduce a tradeoff between exploiting the information from previous tasks and exploring the current task. By applying the approach on Value-At-Risk (and CVAR), the authors show that their algorithm is invariant to scaling and vertical shifting of the blackbox function, and robust towards the presence of harmful previous tasks.
The authors show numerical experiments on synthetic data and real-world data.

**Strengths:**

The paper contains several key contributions that will be helpful for the future works in BO for risk measures.
Major strengths:
  - the paper is very well written and the motivations are very clear: exploit previous tasks knowledge (achieved using V-UCB) to get better BO for risk measures. The literature review is well detailed and the authors explain clearly advantages and drawbacks of each of them.
  -  the construction of the no-regret query sets is elegant and the theorem ensuring that any strategy picking input points in such sets is key contribution in this specific research area in my opinion.
  - the definition of the importance weights exploits well the knowledge from the previous tasks. This also enables to avoid the weight decay to $0$ which is used in the state-of-the-art of meta-BO. The strategy keeps using the knowledge from previous tasks over time.
  - the authors provide theoretical results ensuring that the presented meta-BO of risk measures maintains invariance properties in scaling and shifting of the blackbox function.
  - the numerical experiments show that the approach is robust towards the presence of harmful previous tasks (versus V-UCB).

**Weaknesses:**

Although the paper is very clear and provide important results for the community, there are couple of points that are not completely clear to me.

Major comments/questions:
   - the distribution of Z is supposed to be known. Is it a parameterized distribution that is fitted in practice?
   - the point which is unclear to me is the definition of the lacing values. If I understand well, this means that at each iteration the authors pick a value for z (corresponding to the lacing value) and draw an observation $y_t = f(x_t, z) + \varepsilon$. Is this correct? How are those lacing values computed in practice? I think a paragraph is necessary to explain exactly what is done here. Would it be possible to provide a pseudo-code summarizing the main steps of the algorithm?
  - it would have been interesting to see an illustration of the set $\mathcal{Q}_t$ in the numerical experiments and the computed importance weights. And also to see the evolution of this query set over time.  Figure 1 is not easily readable in my opinion.
  - the authors write that "we want to maximize the size of the query sets". I understand this point since it enables to exploit all the previous tasks. However, it seems that there might exist some tasks that do not bring useful information or harmful information. In such case, wouldn't it be more interesting to discard these tasks from the query set? This question is a bit naive, but from a computational perspective, it would make more sense to consider less previous tasks. Is this correct?
  - Maybe I've missed this point, but could you comment the computational cost of the approach?


Minor comments:
  - titles of subsection 2.2 and Section 3 are the same.
  - In my opinion, there is a lack of details about the transfer knowledge from previous tasks in the settings. For a reader who is not familiar with meta learning, it would have been helpful to know precisely (in the settings) what is the "knowledge" you extract from a previous task. In your case, this is the output of V-UCB. This is explained in the introduction, but this would make the reading easier.

**Questions:**

See above.
Is the code publicly available?

---

> ### Author Response · Authors · 2023-11-19
> **Response to Reviewer 4GcU's Comments**
>
> We would like to thank you for acknowledging the quality of our writing, the motivation, the thorough literature review, and the elegance of the no-regret query set. We also value your acknowledgement of the importance of the priority assignment, the theoretical results, and the numerical experiments.
>
> We provide the following clarifications to address your concerns and question.
>
> > + the distribution of Z is supposed to be known. Is it a parameterized distribution that is fitted in practice?
>
>   We adopt this assumption from previous works on BO of risk measures (Cakmak et al., 2020; Nguyen et al., 2021a;b). In practical applications, we believe one should parameterize and fit the distribution of Z to historical data.
>
> > + the point which is unclear to me is the definition of the lacing values. If I understand well, this means that at each iteration the authors pick a value for z (corresponding to the lacing value) and draw an observation $y_t = f(x_t,z) + \epsilon$. Is this correct? How are those lacing values computed in practice? I think a paragraph is necessary to explain exactly what is done here. Would it be possible to provide a pseudo-code summarizing the main steps of the algorithm?
>
>   Your interpretation regarding the selection of z as a lacing value is accurate. Following your suggestion, we have included the pseudocode of the algorithm in Appendix G of the revised paper to facilitate its implementation for the reader. Additionally, a concise discussion on the definition of lacing values is provided in Appendix C.
>
> > + it would have been interesting to see an illustration of the set $\mathcal{Q}_t$ in the numerical experiments and the computed importance weights. And also to see the evolution of this query set over time. Figure 1 is not easily readable in my opinion.
>
>   We have followed your suggestion to include an illustration of the set $\mathcal{Q}_t$ and the computed importance weights over several iterations in Appendix H.1 of the revised paper.
>
> > + the authors write that "we want to maximize the size of the query sets". I understand this point since it enables to exploit all the previous tasks. However, it seems that there might exist some tasks that do not bring useful information or harmful information. In such case, wouldn't it be more interesting to discard these tasks from the query set? This question is a bit naive, but from a computational perspective, it would make more sense to consider less previous tasks. Is this correct?
>
>   Identifying and excluding harmful tasks to improve computational efficiency, especially when dealing with a large number of harmful tasks, is an interesting direction for future exploration. However, this is a challenging task given our current understanding. The difficulty arises from the absence of a concrete measure of a task's harmfulness. Two risk measures may exhibit similarity in certain local regions while diverging in others, a nuance captured by our assignment of priorities to inputs in $\mathcal{Q}_t$.
>
>   On the other hand, the set $\mathcal{Q}_t$ plays a vital role in ensuring the algorithm's robustness against harmful tasks. As illustrated in Figure 6 of the revised paper, where previous task 1 differs from the current task, the reduction in the size of $\mathcal{Q}_t$ diminishes the negative impact of previous task 1 on the optimization of the current task.
>
>
> > + Maybe I've missed this point, but could you comment the computational cost of the approach?
>
>   We have included a discussion on the computational complexity of the proposed algorithm, comparing it to V-UCB (BO of risk measures without utilizing any previous tasks), in Appendix G of the revised paper. In summary, our algorithm introduces only an additional linear dependence on the number of previous tasks.

---

> ### Author Response · Authors · 2023-11-19
> **Response to Reviewer 4GcU's Comments (continued)**
>
> > Minor comments:
> > + titles of subsection 2.2 and Section 3 are the same.
>
>   We have revised the titles of Subsection 2.2 and Section 3 to be 'Meta-BO of Risk Measures' and 'Our Proposed Solution to Meta-BO of Risk Measures', respectively.
>
> > + In my opinion, there is a lack of details about the transfer knowledge from previous tasks in the settings. For a reader who is not familiar with meta learning, it would have been helpful to know precisely (in the settings) what is the "knowledge" you extract from a previous task. In your case, this is the output of V-UCB. This is explained in the introduction, but this would make the reading easier.
>
>   As you have noticed, in the second paragraph of the introduction (Section 1), we provide a brief overview of the goal of transferring optimization outcomes from previous tasks, such as function evaluation rankings, search space reduction, or the maximizer distribution in the existing literature. However, there is a lack of consensus on the specific knowledge to be transferred from previous tasks to the current one. In our research, we introduce the concept of transferring the local maximizers, which is distinct from existing literature. We will revise to make the discussion of the transfer knowledge clearer in the revised paper.
>
> > + Will the code be publically available?
>
>   We have attached the code in the supplementary materials.
>
> ---
>
> We sincerely hope that the inclusion of pseudocode, supplementary illustrations, and the responses provided above will address any concerns you may have regarding our paper. Your suggestions will be meticulously taken into consideration during the paper's revision process.

---

> > ### Comment · Reviewer_ZY9Q · 2023-11-22
> > **Thank you for the responses**
> >
> > I thank the authors for answering my questions and addressing my comments.
> > I keep my score.

---

> > > ### Author Response · Authors · 2023-11-23
> > > **Thank You for Your Continued Positive Support**
> > >
> > > We are delighted that the our responses have addressed your questions, and we appreciate your continued positive support for our paper.
> > > Once again, thank you for the time and effort you invested in reviewing our paper and our response.

---

### Official Review · Reviewer_4GcU · 2023-11-01

**Soundness:** 2 fair
**Presentation:** 1 poor
**Contribution:** 2 fair
**Rating:** 6
**Confidence:** 3

**Summary:**

The paper proposes a robust meta-BO for risk measures algorithm. It achieves the robustness to harmful previous tasks essentially by constructing the no-regret query set (NQS) that guarantees the convergence of the optimization while using the counting of probably local maximizer in related tasks as the priority within NQS to guide the acquisition. The paper offers theoretical justification for constructing NQS and the regret analysis.

**Strengths:**

1. The construction of NQS guarantees the robustness of the meta-BO.

2. The algorithm is generally well-motivated. Since the key concepts are generally constructed on top of confidence intervals for the risk measure, they bear good interpretability.

**Weaknesses:**

1. It is unclear why the $\lambda$ and $\eta$ are proposed to trade off the goals of NQS explicitly. The algorithm, by default, sets $\lambda=0$ and $\eta=1$ as discussed following the theorem 3.7.

2. The priority mechanism in eq (10) is proposed to incorporate information from previous tasks, but it is actually not robust to harmful previous tasks. This is reflected in both the comparison between the results from theorem 3.2 and previous works' regret bounds and the empirical results.

3. Figure 3 demonstrates the effectiveness of the priority mechanism. Yet since the priority function relies on counting, it is unclear whether the proposed method is robust to the increase of the portion of harmful tasks in $\tau$.

**Questions:**

1. It seems that the construction of NQS basically relies on the upper and lower confidence bounds of the risk measure $\rho_f$ in eq (3). In ordinary BO, the UCB and LCB of the unknown objective function f are the direct equivalence of these confidence bounds for risk measure when the goal is optimized f. Then, an NQS for ordinary meta-BO could be constructed in the same way and could incur a similar regret guarantee. Could the author comment on this extension of NQS in classic meta-BO?

2. The layout is intense and could be more reader-friendly. Could the author reduce the discussion over literature in sections 1 and 2 and leave more room for the essential equations, especially those for the assumptions?

---

> ### Author Response · Authors · 2023-11-19
> **Response to Reviewer 4GcU's Weaknesses**
>
> We appreciate your acknowledgment of our novel development of NQS, the interpretability and motivation of our proposed algorithm. In addressing your inquiries, we would like to offer the following clarifications.
>
> > 1. It is unclear why $\lambda$ and $\eta$ are proposed to trade off the goals of NQS explicitly. The algorithm, by default, sets $\lambda = 0$ and $\eta = 1$ as discussed following the theorem 3.7.
>
>    To assist readers in grasping the essence of our technical approach, we begin by presenting it in a way that starts with basic conceptual ideas accessible to layman readers and then gradually progresses to a rigorous mathematical formulation in Section 3.1. The parameters $\lambda$ and $\eta$ play a crucial role in translating conditions (C1) and (C2) into overlapping and uncertainty conditions. Subsequently, we elucidate the rationale behind choosing $\lambda = 0$ and $\eta = 1$ after a more in-depth discussion on the regret incurred by our proposed algorithm. In contrast, setting them directly and specifically to $\lambda = 0$ and $\eta = 1$ right at the beginning would potentially raise questions regarding alternative formulations of the overlapping and uncertainty conditions (with $\lambda \neq 0$ and $\eta \neq 1$). Hence, we chose to discuss their roles and trade-off first before setting them to specific values. Furthermore, it could be challenging to discuss the trade-off between exploiting previous tasks and exploring current tasks and the size of NQS without parameters $\lambda$ and $\eta$.
>
>
> > 2. The priority mechanism in eq (10) is proposed to incorporate information from previous tasks, but it is actually not robust to harmful previous tasks. This is reflected in both the comparison between the results from theorem 3.2 and previous works' regret bounds and the empirical results.
>
>    The robustness to harmful previous tasks is demonstrated by the algorithm's cumulative regret being sublinear (in contrast to worse than sublinear), even in the presence of harmful previous tasks. This interpretation also aligns with the approach taken in the work of Dai et al. (2022). Moreover, in all existing meta-BO studies, it is expected that the algorithm's performance may deteriorate in the presence of harmful tasks. The critical characteristic here is that given a sufficient number of observations, the algorithm can effectively identify the optimal solution (in contrast to not) irrespective of harmful prior tasks, as indicated by the sublinear cumulative regret. As far as we are aware, this particular feature is not found in the majority of meta-BO research, except for the notable work of Dai et al. (2022). Consequently, we consider demonstrating this property for our proposed algorithm to be a significant contribution.
>
>
> > 3. Figure 3 demonstrates the effectiveness of the priority mechanism. Yet since the priority function relies on counting, it is unclear whether the proposed method is robust to the increase of the portion of harmful tasks in $\tau$.
>
>    The robustness does not come from the priority assignment; rather, it mainly arises from the construction of NQS and the theoretical guarantee that choosing any input from NQS leads to a no-regret algorithm, which is a unique contribution of our work. Additionally, we have included a visualization of the algorithm in the scenario where the set of previous tasks consists solely of harmful tasks, as illustrated in Figure 6 of the revised paper.
>
>    In Figure 2, note that the sets $\\mathcal{T}\_{\\texttt{harmful-neg-scale}}$ and $\\mathcal{T}\_{\\texttt{harmful-hshift}}$ of previous tasks consist of all harmful tasks (i.e., no useful previous tasks). Hence, we have empirically demonstrated the performance of our algorithm when the portion of harmful tasks is $100\%$ apart from the sublinear cumulative regret proof. This is highlighted in the last paragraph of Section 4: "In the extreme case when $\mathcal{T}$ contains only harmful tasks ...".

---

> ### Author Response · Authors · 2023-11-19
> **Response to Reviewer 4GcU's Questions**
>
> > 1. It seems that the construction of NQS basically relies on the upper and lower confidence bounds of the risk measure $\rho_f$ in eq (3). In ordinary BO, the UCB and LCB of the unknown objective function f are the direct equivalence of these confidence bounds for risk measure when the goal is optimized f. Then, an NQS for ordinary meta-BO could be constructed in the same way and could incur a similar regret guarantee. Could the author comment on this extension of NQS in classic meta-BO?
>
> This is a very insightful observation! As outlined by the reviewer, we think there is potential to extend NQS to classic meta-BO by substituting the upper and lower confidence bounds of the risk measure with UCB and LCB of the unknown objective function. We will provide a comprehensive discussion of this extension in the revised paper to enhance the impact of our work within the BO community. In the context of risk measures, where a solution for meta-BO incorporating risk measures is lacking, we consider it innovative and practical to position and define the scope of our work as being the first work in the domain of meta-BO focused on risk measures.
>
>
> > 2. The layout is intense and could be more reader-friendly. Could the author reduce the discussion over literature in sections 1 and 2 and leave more room for the essential equations, especially those for the assumptions?
>
> Given that our work is situated within the domains of BO, meta-BO, and BO of risk measures, we agree with the reviewer's observation about the extensive discussion on the literature and background. However, we also believe that it is crucial to present this background information to ensure readers can fully grasp the equations and assumptions. If the reviewer has suggestions for improving the discussion of any specific assumptions or equations in mind, we would greatly appreciate them. We are committed to enhancing the clarity and coherence of these sections in the revised paper based on any feedback provided.
>
> ---
>
> Once again, thank you for your insightful suggestions, which we will integrate into our revised paper. We highly appreciate your questions and hope that our responses contribute to enhancing your perception of our paper.

---

> > ### Author Response · Authors · 2023-11-23
> > **Appreciating Your Reviews and Humbly Ask For Feedback**
> >
> > Dear Reviewer,
> >
> > During the remaining hours of the author-reviewer discussion period, it would be great if you could inform us whether our response has addressed your concerns regarding our paper. Your dedication to reviewing our work despite your busy schedule is genuinely appreciated. Lastly, we just want to say thank you for your evaluation of both our paper and our rebuttal.
> >
> > Kind regards,
> >
> > Authors

---

> > > ### Comment · Reviewer_4GcU · 2023-11-23
> > >
> > > Dear Authors,
> > >
> > > I've read the rebuttal and my fellow reviewers' comments carefully. I appreciate the author's clarification. I stand by my original comments that the regret analysis and the priority mechanism are reasonable yet won't outperform the existing method in the worst-case scenario.  The NQS is an interesting mechanism that could be of broader interest other than the risk measures. The efficiency and robustness against model-misspecification of the construction of NQS might incur additional concerns over the practicality. But in general, the corresponding theoretical analysis was fine. I'm not inclined to veto the work.
> > >
> > > Thanks.

---

> > > > ### Author Response · Authors · 2023-11-23
> > > > **Thank You for Your Valuable Feedback**
> > > >
> > > > Dear Reviewer,
> > > >
> > > > We sincerely appreciate your prompt response and the increased score which hold great significance for us. We highly appreciate the commitment you have shown in carefully examining all the feedback and assessing our paper.
> > > >
> > > > Kind regards,
> > > >
> > > > Authors

---

### Official Review · Reviewer_25FF · 2023-11-13

**Soundness:** 3 good
**Presentation:** 3 good
**Contribution:** 3 good
**Rating:** 6
**Confidence:** 3

**Summary:**

This work considers how to leverage prior knowledge when performing Bayesian optimization of risk measures (VaR, CVaR). Prior works in the literature either consider the availability of prior knowledge to warm-start BO for non-risk measures, or optimization of risk measures without using prior knowledge. This work tackles both of these issues simultaneously. The key novelty of the work is the development of a "no-regret query set" that defines a set of input query sequences that would result in sub-linear cumulative regret. This set contains those inputs that are likely to be maximizers of the risk measure given the posterior belief about the underlying function and also provides additional information toward the estimation of risk measures. Once such a set is established, prior knowledge can be used to rank/prioritize the query sequences. As any query sequence within this query set would result in sublinear cumulative regret, even if the prior tasks are very different, asymptotic guarantees are still preserved (although finite sample rates deteriorate).

**Strengths:**

S1. The idea of a no-regret-query set is novel, to the best of my knowledge.

S2. Authors carefully consider the impact of the quality of knowledge from prior tasks and how those influence the final results.

**Weaknesses:**

W1. Empirical results are on extremely toy examples.

W2. Like prior works, this work is also limited in terms of assuming exact knowledge of the noise distribution. Seems like a strong requirement.

W3. Not really a weakness, but my guess is that this paper might have a much bigger audience at other venues (e.g., ICML/AISTATS).

**Questions:**

1. The key contribution seems to be the introduction of the idea of a no-regret-query set (NQS). Maybe I missed it, but it feels like it is more generally applicable than the exact problem being considered in this work. If not, perhaps it would be beneficial to know what makes NQS restricted to the setting of risk measures? Also, seems like various other side-information can be used to prioritize samples within NQS.

2. While I understand that the requirement of noise distribution is similar to assumptions made in the prior work and is not directly related to the proposed method, however, since the paper currently is phrased as a solution for optimizing risk measures, it would be useful for the readers to have the assumption formally stated in Section 2.

3. Can the authors provide some experiments on the scalability of the proposed approach? The toy examples considered all have dimensions less than 5 or 6.

4. I think the introduction can be compressed significantly. Currently, the main contribution of the work starts at the end of page 5.

5.  If \lambda goes near 0, then \eta can go near infinity and the regret guarantee becomes vacuous. This results in two important hyper-parameters of the problem: \eta and \lambda, that would essentially control exploration-exploitation. While 3.3 gives one recommendation, how sensitive are the practical results to the choice of their values?

6. I do not understand the Lacing value choice of the noise variable, and why is important to obtain the desired result. Given that Theorem 3.2 is the core contribution of the work, I think it would be imperative to explain this assumption in more detail.

---

> ### Author Response · Authors · 2023-11-19
> **Response to Reviewer 25FF's Questions**
>
> We greatly appreciate your recognition of our innovative development of the no-regret query set and the careful consideration of the quality of knowledge from prior tasks to the current task.
>
> In the following paragraphs, we would like to address your questions and concerns.
>
> > 1. The key contribution seems to be the introduction of the idea of a no-regret-query set (NQS). Maybe I missed it, but it feels like it is more generally applicable than the exact problem being considered in this work. If not, perhaps it would be beneficial to know what makes NQS restricted to the setting of risk measures? Also, seems like various other side-information can be used to prioritize samples within NQS.
>
> This is an excellent observation! Our original aim was to tackle the unexplored domain of meta-learning for BO of risk measures. This led to the introduction of the novel NQS. From the insightful comment of the reviewer, we acknowledge that this approach proves to be more versatile than initially anticipated. At present, we recognize the potential application of the NQS technique to various existing BO variants within the realm of meta-learning, including:
>
> + Contextual Gaussian process bandit optimization (Krause et al., 2011).
> + Adversarially robust optimization with Gaussian processes (Bogunovic et al., 2018).
>
> We will incorporate this expanded discussion in the revised version of the paper.
>
> > 2. While I understand that the requirement of noise distribution is similar to assumptions made in the prior work and is not directly related to the proposed method, however, since the paper currently is phrased as a solution for optimizing risk measures, it would be useful for the readers to have the assumption formally stated in Section 2.
>
> To enhance clarity, we plan to restate the assumption outlined in the introduction when we delve into Section 2.
>
> > 3. Can the authors provide some experiments on the scalability of the proposed approach? The toy examples considered all have dimensions less than 5 or 6.
>
> We have incorporated experiments involving 9-dimensional inputs in Appendix H.2 of the revised paper. The outcomes for both VaR and CVaR risk measures, utilizing various sets of previous tasks (including harmful and useful tasks), are presented in Figure 8 of the revised paper. These results are consistent with the findings from other experiments detailed in the paper.
>
> We would like to highlight our experiments with the yacht hydrodynamics and portfolio optimization datasets which makes our experiments section comparable with that of existing works on BO of risk measures.
> The difficulty in finding real-world experiments arises the absence of existing meta-BO research that considers risk measures. This results in a lack of previous real-world tasks within the meta-BO literature that incorporate real-world environmental random variables.
>
> Additionally, we enhance the overall comprehensiveness of our experiments section by including the CVaR experimental results in Appendix H of the revised paper.
>
> It is worth mentioning that research on high-dimensional BO is ongoing. One feasible approach to address this challenge is the use of simple techniques, such as random projection, to effectively reduce the input dimension of the optimization problem.
>
> > 4. I think the introduction can be compressed significantly. Currently, the main contribution of the work starts at the end of page 5.
>
> As our research is positioned in the domains of BO, meta-BO, and BO of risk measures, we agree with the reviewer about the comprehensive introduction which aims to discuss the literature and background in these 3 domains. Although the technical solution is on page 5, we have attempted to summarize the contribution in the last paragraph of the introduction (Section 1). This serves to provide readers with a preview of the paper's essence before delving into the background, the problem statement, and the technical solution.

---

> ### Author Response · Authors · 2023-11-19
> **Response to Reviewer 25FF's Questions (continued)**
>
> > 5. If \lambda goes near 0, then \eta can go near infinity and the regret guarantee becomes vacuous. This results in two important hyper-parameters of the problem: \eta and \lambda, that would essentially control exploration-exploitation. While 3.3 gives one recommendation, how sensitive are the practical results to the choice of their values?
>
>    We would like to thank the reviewer for the insightful question. While $\eta$ can go near infinity when $\lambda$ goes near $0$, from equation (7), note that the LHS is bounded from below by $\min_{x' \in \mathcal{X}} \rho_{u_t}(x';\alpha) - \rho_{l_t}(x';\alpha) > 0$ (since the posterior standard deviation is strictly positive). Therefore, we will revise the upper bound of $\eta$ as $\eta \le \min(1/\lambda, (\rho_{u_t}(x_t^+;\alpha) - \rho_{l_t}(x_t^-;\alpha)) / (\min_{x' \in \mathcal{X}} \rho_{u_t}(x';\alpha) - \rho_{l_t}(x';\alpha)))$. As a result, when $\lambda$ goes to $0$, $\eta$ is still finite and the regret guarantee is still meaningful.
>
>    In practical applications, we suggest setting $\lambda$ to 0 and $\eta$ to 1 when practitioners believe that previous tasks are useful for the current task. Conversely, if practitioners are risk-averse and believe that previous tasks are predominantly harmful to the current task, they can set $\lambda$ close to 1 and $\eta$ to 1. This adjustment makes the algorithm more akin to GP-UCB, which reduces the use of information from previous tasks. This corresponds to our discussion on the exploration-exploitation trade-off in the paragraph following Lemma 3.1.
>
> > 6. I do not understand the Lacing value choice of the noise variable, and why is important to obtain the desired result. Given that Theorem 3.2 is the core contribution of the work, I think it would be imperative to explain this assumption in more detail.
>
>    Given our focus on meta-BO for risk measures, it is essential, at each BO iteration, to query both $x_t$ and $z_t$ as detailed in Section 2.1, an approach that is adopted from existing literature on BO of risk measures. The transfer of information from BO solutions of prior tasks to the current one primarily occurs through the selection of $x_t$. Selecting for specific values for $z_t$ ensures the no-regret property, akin to previous studies on BO of risk measures. The absence of a specific choice for $z_t$ could compromise the guarantee of sublinear regret, regardless of the chosen $x_t$.
>
> ---
>
> We sincerely hope that the additional experimental results and our above clarifications can resolve your concerns regarding our paper and improve your opinion of our work. We will carefully incorporate your suggestions into the revised paper.

---

> > ### Author Response · Authors · 2023-11-23
> > **Appreciating Your Reviews and Humbly Ask For Feedback**
> >
> > Dear Reviewer,
> >
> > We just want to express our appreciation for your assessment of both our paper and our rebuttal. If time allows during the remaining author-reviewer discussion period, we would appreciate your feedback on whether our response has sufficiently addressed your concerns about our paper. Thank you once more for your valuable insights and suggestions during the rebuttal process.
> >
> > Kind regards,
> >
> > Authors

---

> > > ### Comment · Reviewer_25FF · 2023-11-23
> > >
> > > Thank you for your efforts towards the rebuttal! I would like to stay with my original score.

---

> > > > ### Author Response · Authors · 2023-11-23
> > > > **Thank You for Your Feedback and Review of Our Paper**
> > > >
> > > > Thank you for your timely response and for reviewing our paper.
> > > > We sincerely value the time and effort you dedicated to this extensive process.

---

### Meta-Review · Area_Chair_LEkK · 2023-12-12

**Metareview:**

All reviewers agree that this is a decent contribution to the literature on Bayesian optimization and shows how to incoprorate prior knowledge when optimizing risk scores. The paper offers new methodological insights.

**Justification For Why Not Higher Score:**

The scope of the work could have limited target audience. The experimental evaluation was mostly on toy examples.

**Justification For Why Not Lower Score:**

The paper offers a new methodological advance in Bayesian optimization

---

### Decision · Program_Chairs · 2024-01-16

Accept (poster)